# Semi-knockoffs: a model-agnostic Conditional Independence Testing method with finite-sample guarantees

Angel Reyero Lobo [1 2]   Bertrand Thirion [2]   Pierre Neuvial [1]

## Abstract

Conditional independence testing (CIT) is essential for reliable scientific discovery. It prevents spurious findings and enables controlled feature selection. Recent CIT methods have used machine learning (ML) models as surrogates of the underlying distribution. However, model-agnostic approaches require a train-test split, which reduces statistical power. We introduce *Semi-knockoffs*, a CIT method that can accommodate any pre-trained model, avoids this split, and provides valid p-values and false discovery rate (FDR) control for high-dimensional settings. Unlike methods that rely on the model-$X$ assumption (known input distribution), Semi-knockoffs only require conditional expectations for continuous variables. This makes the procedure less restrictive and more practical for machine learning integration. To ensure validity when estimating these expectations, we present two new theoretical results of independent interest: (i) stability for regularized models trained with a null feature and (ii) the double-robustness property.

## 1. Introduction

Scientific discovery involves identifying features relevant to a target. For example, given a set of genes, one may wish to find those important for a particular disease. In high-dimensional settings, some variables may appear important only because they are correlated with truly relevant ones. To address this, the problem is formalized as a conditional independence testing (CIT) task (Candès et al., 2018), where a variable is important if it provides information about the target beyond what is contained in the other features. Under the standard assumption that no covariate is exactly a function of the others (Candès et al., 2018; Verdinelli & Wasserman, 2024), this set of important variables is unique and known as the Markov blanket.

In recent years, machine learning (ML) models have significantly improved predictive performance across complex tasks, potentially providing useful insights for scientific discovery (Ewald et al., 2024). However, their increased expressiveness often comes at the cost of interpretability, making it difficult to evaluate the relevance of individual features. Addressing this challenge is the focus of interpretable ML, particularly global variable importance (Covert et al., 2020; Williamson et al., 2023; Molnar, 2025). This lack of transparency widens the gap between assigning predictive relevance in a model (variable importance) and identifying the Markov blanket (variable selection) (Reyero-Lobo et al., 2025b). Most variable importance methods are heuristic and abstract away from the underlying distribution, making rigorous statistical guarantees challenging and often asymptotic (Williamson et al., 2023; Verdinelli & Wasserman, 2024). In contrast, controlled variable selection methods can provide strong guarantees, such as finite-sample type-I error or false discovery rate (FDR) control, but they typically rely on transparent models (e.g., Lasso in knockoffs or CRT; Candès et al. (2018)) or on linear correlation (e.g., GCM; Shah & Peters (2018) or dCRT; Liu et al. (2022)).

When procedures are based on complex models, quantifying the impact of a single feature becomes nontrivial, unlike in linear models where coefficients provide a natural measure of importance. Many approaches assess the drop in predictive performance when either the model is refitted without the feature (Lei et al., 2018; Williamson et al., 2021) or by conditionally replacing it (Strobl et al., 2008; Chamma et al., 2024). Several methods based on the model-X framework (Candès et al., 2018) combine these ideas with valid type-I error in a model-agnostic manner (Tansey et al., 2022). However, they typically rely on symmetry between the original and an artificial null distribution, requiring evaluation on an independent test set. This requires a train-test split, which can limit performance in small-sample regimes compared to simpler models (Liu et al., 2022). Knockoffs are

[1]Univ Toulouse, INSA Toulouse, CNRS, IMT, Toulouse, France [2]Université Paris-Saclay, Inria, CEA, Palaiseau, 91120, France. Correspondence to: Angel Reyero Lobo <angel.reyero-lobo@inria.fr>.

*Proceedings of the 43rd International Conference on Machine Learning*, Seoul, South Korea. PMLR 306, 2026. Copyright 2026 by the author(s).

particularly popular in high-dimensional settings due to their computational efficiency and finite-sample FDR control. These procedures construct knockoff variables, define suitable statistics, and apply a data-dependent threshold. However, designing statistics based on pre-trained models remains an open challenge: existing attempts fail to fully comply with the knockoff framework or provide only asymptotic type-I error (Watson & Wright, 2021). Moreover, sampling knockoff variables is challenging and raises concerns regarding finite-sample validity (Blain et al., 2025).

Our main contributions are summarized as follows:

- We introduce *Semi-knockoffs*, a method designed for controlled variable selection while accommodating arbitrary pre-trained ML models. It does not require a train-test split, nor the construction of exact knockoff variables, and it avoids the constraints imposed on knockoff statistics. The method provides finite-sample control of both type-I error and the false discovery rate.

- We study the control under estimated, rather than oracle samplers. First, we establish a *stability* result for regularized learners, yielding distributional convergence of the Semi-knockoffs. Second, we provide a *double-robustness* result, conjecturing the control as the model and sampler together achieve faster convergence.

- We demonstrate our approach through extensive numerical experiments, drawing on state-of-the-art methods from variable selection and importance.

**Notation:** Let $\{(x_i, y_i)\}_{i=1}^n$ be i.i.d. samples from the joint distribution $\mathcal{L}(X, Y)$, where $X \in \mathcal{X} \subset \mathbb{R}^p$ and $Y \in \mathcal{Y} \subset \mathbb{R}$. We denote by $\widehat{m} : \mathcal{X} \to \mathcal{Y}'$ a predictive model and by $l : \mathcal{Y}' \times \mathcal{Y} \to \mathbb{R}_+$ a loss function. For $j \in [p]$, we refer to the $j$-th coordinate of $X$ as the feature of interest, denoted by $X^j$, and write $X^{-j}$ for $X$ restricted to all but the $j$-th coordinate. We denote by $\mathcal{H}_0 \subseteq [p]$ the null set. For a sequence $\{a_n\}$, $O_{\mathbb{P}}(a_n)$ denotes boundedness in probability at rate $a_n$, while $o_{\mathbb{P}}(a_n)$ denotes convergence to zero in probability at that rate. Any additional notation is introduced as needed and summarized in Appendix A.

## 2. Related work

We first introduce variable importance and then variable selection methods; full definitions are in Appendix B.

### 2.1. Leave One Covariate Out (LOCO)

A first model-agnostic approach consists of measuring the drop in predictive performance when the model is refitted without the coordinate of interest (Lei et al., 2018). LOCO nonparametric efficiency has been carefully studied in Williamson et al. (2021; 2023). However, standard $t$-tests

are not applicable for conditional independence testing, as the variance of the resulting statistic vanishes under the null, invalidating the central limit theorem. To address this, Williamson et al. (2023) propose performing the refitting on an independent dataset, which corrects the variance and enables asymptotic type-I error control. In practice, however, this approach incurs substantial power loss (Reyero-Lobo et al., 2025a). Alternative solutions instead artificially inflate the variance by adding a correction term of order $O_{\mathbb{P}}(n^{-1/2})$ and then apply Chebyshev's inequality for asymptotic type-I error control (Verdinelli & Wasserman, 2024).

### 2.2. Conditional Feature Importance (CFI)

Since refitting can be costly, early heuristics aimed to remove the information carried by a feature without changing the model. These methods evaluate performance on test data in which the feature's information is disrupted. A classical strategy is to break the dependence marginally: predictions are made on observations where all coordinates are preserved except the $j$-th, which is permuted. This yields the MDA (Breiman, 2001) or PFI (Mi et al., 2021).

Despite its simplicity, PFI has extrapolation issues because it forces the model to make predictions in low-density regions of the input space (Hooker & Mentch, 2019). To mitigate this issue, several studies have proposed *conditional* permutation, which preserves the dependence structure with the remaining inputs while breaking the association with the target variable (Strobl et al., 2008). This approach has demonstrated robust empirical performance (Chamma et al., 2024) and possesses a double robustness property (Reyero-Lobo et al., 2025a), implying that a good conditional sampler or model is sufficient to detect null features.

Nonetheless, providing statistical guarantees remains difficult. Reyero-Lobo et al. (2025a) show that standard $t$-tests are invalid due to vanishing variance (as in LOCO). To address this, they either inflate the variance by an $O(n^{-1/2})$ term (or $O(n^{-a})$ for $a < 2$ in the linear case) and apply Markov's inequality for asymptotic type-I control, or use a Wilcoxon test. However, this requires an independent dataset to avoid overfitting, enforcing a train-test split.

### 2.3. Model-X variable selection methods

Candès et al. (2018) introduced the model-X framework, which assumes that the input distribution is known (or can be easily estimated), enabling conditional sampling. This is reasonable when the covariates can be experimentally controlled (e.g., clinical trials), when modeling the input is easier than modeling the input-output association (e.g., genomics), or when abundant unlabeled data are available.

### 2.3.1. CONDITIONAL RANDOMIZATION TEST (CRT)

The CRT computes a test statistic $T(X^j, X^{-j}, y)$ that measures the importance of feature $j$—for example, a Lasso coefficient—and compares it to its values obtained on conditionally resampled datasets. Under the model-X assumption, one can sample $\widetilde{X}^j \sim \mathcal{L}(X^j \mid X^{-j})$. Under the null, the pairs $T(X^j, X^{-j}, y)$ and $T(\widetilde{X}^j, X^{-j}, y)$ are exchangeable, which allows an exact finite-sample type-I error control.

The original CRT of Candès et al. (2018) is computationally prohibitive, as it requires numerous refittings of the predictive model. To alleviate this, Tansey et al. (2022) proposed the Holdout Randomization Test (HRT), a model-agnostic approach that evaluates a single global model across multiple conditional resamplings. However, to preserve the exchangeability of performance scores, the model must be evaluated using data that is independent of the training set. This necessitates a train-test split, which results in a loss of power.

To mitigate this limitation, Liu et al. (2022) introduced the distilled CRT (dCRT), which performs a *feature-wise* rather than *sample-wise* split. Instead of refitting a full model for each feature and each resample, the dCRT performs a single expensive step per feature: a regression of $y$ on $X^{-j}$ using any predictive model. The residual represents the part of $y$ unexplained by the other features, and the test examines whether $X^j$ explains additional variation in this residual. However, while the distillation step is model-agnostic, the final association test between $X^j$ and $y$ is not.

### 2.3.2. KNOCKOFFS

Knockoffs were introduced by Barber & Candès (2015) and generalized to the model-X framework by Candès et al. (2018). They have since become popular for variable selection, providing FDR control without relying on $p$-values or dependence assumptions such as PRDS (Benjamini & Hochberg, 1995). It relies on three components: knockoff variables $\widetilde{X} \in \mathbb{R}^p$, knockoff statistics $W \in \mathbb{R}^p$, and a data-dependent threshold (see Appendix B.3 for details).

Briefly, knockoff variables act as synthetic copies of $X$ and allow one to distinguish genuine from spurious importance. Their construction is more restrictive than that of conditional samplers used in the CRT or CFI: they must satisfy a joint *pairwise exchangeability* for the entire vector $(X, \widetilde{X})$, rather than only matching $\mathcal{L}(X^j \mid X^{-j})$. The knockoff statistic $W^j$ must satisfy an antisymmetry property: swapping the $j$-th feature with its knockoff must flip the sign of $W^j$ and remain invariant when swapping other features. Finally, to control the FDR at level $q$, the threshold is given by

$$T_q = \min\left\{ t \in \mathcal{W} : \frac{1 + \#\{j : W^j \leq -t\}}{\#\{j : W^j \geq t\} \vee 1} \leq q \right\}. \quad (1)$$

The FDR control of the knockoffs relies fundamentally on a symmetry of the statistics under the null. This symmetry underpins the validity of the knockoff threshold, which exploits the balance between positive and negative statistics. Formally, Lemma 1 of Candès et al. (2018) shows that:

**Lemma 2.1** (exchangeability)**.** *Conditionally on* $(|W_1|, \dots, |W_p|)$, *the signs of* $W_j$ *for* $j \in \mathcal{H}_0$ , *are i.i.d. coin flips.*

However, achieving such symmetry with black-box models is nontrivial. For example, comparing the performance of a model $\widehat{m}$ on the original data versus a distribution in which the $j$-th coordinate is replaced by its knockoff (Watson & Wright, 2021) fails to satisfy the required exchangeability. The difficulty stems from the fact that this statistic is not antisymmetric: swapping a feature with its knockoff in a coordinate other than $j$ does not leave the statistic invariant.

## 3. Oracle Semi-knockoffs

The main distinction between model-agnostic methods (e.g., CFI, HRT) and coefficient-based approaches (e.g., LCD knockoffs, dCRT) is that the latter yield direct and interpretable measures of feature relevance through estimated coefficients. In contrast, model-agnostic methods must infer relevance indirectly, typically by comparing losses on original versus perturbed data, i.e., via quantities such as

$$l(\widehat{m}(\widetilde{X}^{(j)}), y) - l(\widehat{m}(X), y),$$

where $\widetilde{X}^{(j)}$ is equal to $X$ except for the $j$-th coordinate, which is conditionally resampled ($\widetilde{X}^{(j)j} \sim \mathcal{L}(X^j \mid X^{-j})$).

We observe that independence between the test set and the model is required; otherwise, the test would be biased, since the two distributions $l(\widehat{m}(\widetilde{X}^{(j)}), y)$ and $l(\widehat{m}(X), y)$ would not be the same under the null.

A first possibility to avoid this data-splitting requirement is to sample on both sides from the conditional distribution, and thus to base the statistic on

$$l(\widehat{m}(\widetilde{X}_1^{(j)}), y) - l(\widehat{m}(\widetilde{X}_2^{(j)}), y),$$

where both $\widetilde{X}_1^{(j)}$ and $\widetilde{X}_2^{(j)}$ differ only in the $j$-th feature, which is independently sampled conditional on the remaining features, or on the remaining features together with the output, respectively, i.e.,

$$\widetilde{X}_1^{j(j)} \sim \mathcal{L}(X^j \mid X^{-j}) \quad \text{and} \quad \widetilde{X}_2^{j(j)} \sim \mathcal{L}(X^j \mid X^{-j}, y).$$

By doing so, we break the symmetry under the alternative while preserving it under the null. Under the null, since $X^j \perp\!\!\!\perp y \mid X^{-j}$, both samples are drawn independently from the same distribution, ensuring exchangeability. Under the alternative, the distributions differ because $y$ provides

additional information about $X^j$. Note that imputing a feature while conditioning on $y$ has been shown to be effective for handling missing values (D'Agostino McGowan et al., 2024).

However, since sampling from these conditional distributions is generally challenging, a simpler statistic can be obtained using a regression-based conditional sampler, as in Fisher et al. (2019); Chamma et al. (2024). Specifically, they regress the $j$-th feature on the others and sample from the residuals. We proceed similarly, but also include in the input of the regression the target $y$. Formally, we are interested in

$$\nu_j(X^{-j}) = \mathbb{E}[X^j | X^{-j}] \text{ and } \rho_j(X^{-j}, y) = \mathbb{E}[X^j | X^{-j}, y].$$

We drop the subscript $j$ when clear from context. Note that both quantities can be estimated with ML models.

Then, the new $j$-th coordinate of $\widetilde{X}_1^{(j)}$ is not sampled directly from $\mathcal{L}(X^j | X^{-j})$, but rather defined as

$$\widetilde{X}_1^{(j)j} = \nu(X^{-j}) + \{x_l^j - \nu(x_l^{-j})\} = \nu(X^{-j}) + \epsilon_{j,1}^l,$$

and for $\widetilde{X}_2^{(j)}$, instead of sampling from $\mathcal{L}(X^j | X^{-j}, y)$,

$$\widetilde{X}_2^{(j)j} = \rho(X^{-j}, y) + \{x_k^j - \rho(x_k^{-j}, y_k)\} = \rho(X^{-j}, y) + \epsilon_{j,2}^k,$$

where $x_l$ and $(x_k, y_k)$ are sampled independently (Alg. 2), so that $\epsilon_{j,1}^l$ and $\epsilon_{j,2}^k$ are their respective theoretical residuals for predicting $x_l^j$ and $x_k^j$.

*Remark* 3.1. We emphasize that this does not produce valid knockoff variables. Obtaining valid knockoffs would require sequential conditional sampling based on prior regressions (Candès et al., 2018) and assumptions on the underlying distribution (e.g., Gaussian designs (Blain et al., 2025)). Importantly, such conditions are not required for semi-knockoff validity.

*Remark* 3.2. This concerns continuous features. For categorical ones, a simple logistic regression can be used to estimate the conditional class probabilities, and all arguments and below theory extend directly to this setting.

Thus, under the null, we still have exchangeability since

$$\rho_j(X^{-j}, y) = \mathbb{E}[X^j | X^{-j}, y] = \mathbb{E}[X^j | X^{-j}] = \nu_j(X^{-j}).$$

Hence, under the null we sample independently from the same distribution. Under the alternative, however, $y$ provides additional information about the $j$-th coordinate, leading to more accurate predictions and thus less perturbation.

Finally, most procedures are tailored to either $p$-value computation or FDR control, and transitioning between the two is nontrivial. While $p$-values can in principle be converted to FDR control via (Benjamini & Hochberg, 1995), the required dependence assumptions are rarely checkable, and attempts in the reverse direction often yield pseudo–$p$-values

lacking theoretical guarantees (Nguyen et al., 2020). In the next sections, we introduce two versions of Semi-knockoffs that rely on the same exchangeability principle and provide either valid $p$-values or FDR control. The corresponding pseudocode is deferred to Appendix C.

### 3.1. Type-I error control

Since each coordinate is symmetric around 0 under the null hypothesis and potentially larger under the alternative, it is possible to apply any nonparametric paired test comparing

$$l(\widehat{m}(\widetilde{X}_1^{(j)}), y) \quad \text{and} \quad l(\widehat{m}(\widetilde{X}_2^{(j)}), y),$$

for instance the **sign test** or the **Wilcoxon** test. This yields finite-sample type-I error control for any model $\widehat{m}$.

**Theorem 3.3** (Type-I error). *Given $\nu_j$ and $\rho_j$, nonparametric paired test Semi-knockoffs (Algorithm 2) provide valid $p$-values.*

Note that a t-test is not valid since under the null the variance vanishes and then the asymptotic normality does not hold (Williamson et al., 2023; Verdinelli & Wasserman, 2024).

This approach eliminates the need for a train–test split: rather than requiring that the conditional samples resemble a held-out test set under the null, we rely solely on the symmetry of the sampling mechanism. A further advantage is that, although we implicitly use the CPI conditional sampler (Chamma et al., 2024), which was shown to be valid under an additive-innovation assumption in (Reyero-Lobo et al., 2025a), we do not need to impose such assumptions on the input distribution to obtain semi-knockoff validity.

### 3.2. FDR control

We note that the sign of each Semi-knockoff statistic $W_{\text{SKO}}^j$ (Algorithm 4) is uniformly distributed as $\mathcal{U}(\{\pm 1\})$ and independent of the signs of the other features. This was not the case in the held-out versions, where dependencies remained across features. Hence, the exchangeability condition required for knockoffs (Lemma 2.1) is satisfied. Consequently, we can estimate the set of important features in the same way as for standard knockoffs: we select the threshold $T_q$ as in (1) and select

$$S_{\text{SKO}} = \{j : W_{\text{SKO}}^j \geq T_q\}.$$

This procedure (Algorithm 3) controls the FDR at level $q$:

**Theorem 3.4** (FDR control). *Given $\nu_j$ and $\rho_j$ for every $j$, then $\text{FDR}(S_{\text{SKO}}) \leq q$.*

## 4. Semi-knockoffs

### 4.1. Algorithm

In practice, the conditional expectations $(\nu_j, \rho_j), j \in [p]$ are unknown and must be estimated, for instance using ML

models. Using the same notation as before, we denote by $\widetilde{X}_1'^{(j)}$ (resp. $\widetilde{X}_2'^{(j)}$) the versions that use $\widehat{\nu}_j$ instead of $\nu_j$ (resp. $\widehat{\rho}_j$ instead of $\rho_j$). The practical implementation for type-I error control using the Wilcoxon test is given in Algorithm 1, and the FDR procedure is provided in Appendix 4.

---

**Algorithm 1** Semi-knockoffs Wilcoxon (`SKO-Wcx`)

**Input:** Model $\widehat{m}$, data $\{(X_i, y_i)\}_i^n$, significance level $\alpha$, a feature $j$
    **(Regression 1)** Fit $\widehat{\nu}_j \approx \mathbb{E}[X^j \mid X^{-j}]$
    **(Regression 2)** Fit $\widehat{\rho}_j \approx \mathbb{E}[X^j \mid X^{-j}, y]$
Compute residuals $\widehat{\epsilon}_{j,1} = X^j - \widehat{\nu}_j(X^{-j})$
Compute residuals $\widehat{\epsilon}_{j,2} = X^j - \widehat{\rho}_j(X^{-j}, y)$
Sample $\pi_{j,1}$ and $\pi_{j,2}$ permutations of $\{1, \ldots, n\}$
Construct

$$\widetilde{X}_{1,i}'^{(j)} = \widehat{\nu}_j(X_i^{-j}) + \widehat{\epsilon}_{j,1,\pi_{j,1}(i)}$$
$$\widetilde{X}_{2,i}'^{(j)} = \widehat{\rho}_j(X_i^{-j}, y_i) + \widehat{\epsilon}_{j,2,\pi_{j,2}(i)}$$

Compute Wilcoxon test between

$$\left\{ l\big(\widehat{m}(\widetilde{X}_{1,i}'^{(j)}), y_i\big) \right\} \text{ and } \left\{ l\big(\widehat{m}(\widetilde{X}_{2,i}'^{(j)}), y_i\big) \right\}$$

**Return** the computed p-value

---

At first sight, estimating $\mathbb{E}[X^j \mid X^{-j}, y]$ may seem contradictory, since the model-$X$ framework makes no assumptions about the relationship between $X$ and $y$. However, as discussed in Section 4.2, these regressions do not require complex models, so the procedure remains efficient.

Moreover, we provide two results to study type-I error and FDR under these estimations. The first one (Section 4.3) shows that

$$l(\widehat{m}(\widetilde{X}_1'^{(j)}), y) \quad \text{and} \quad l(\widehat{m}(\widetilde{X}_2'^{(j)}), y)$$

converge in 1-Wasserstein distance under the null hypothesis. Hence, we sample iid from the same distribution. This follows from a stability result (Section 4.2) for $\widehat{\nu}$ and $\widehat{\rho}$, providing a finite-sample bound between a regularized ERM fitted with and without an unimportant coordinate.

The second result (Section 4.5) is that, by extending a double-robustness argument from Reyero-Lobo et al. (2025a) (Section 4.4), the estimated statistic should preserve the sign of the theoretical quantity with high probability.

Finally, note that perturbation-based methods often suffer from extrapolation issues (PFI, Hooker & Mentch (2019)), since predictions are made far from the training distribution. We do not expect this issue to arise here, as we condition on the remaining coordinates, and this sampling strategy has already shown strong performance (Chamma et al., 2024).

## 4.2. Null feature stability of the ERM

Under the model-X assumption, estimating $\nu_j$ is straightforward because the relationship between $X^j$ and $X^{-j}$ is assumed to be simple. For instance, when $X \sim \mathcal{N}(\mu, \Sigma)$ is Gaussian, a linear model is sufficient since

$$\nu_j(x^{-j}) = \mathbb{E}[X^j | X^{-j}] = \mu^j + \Sigma_{j,-j} \Sigma_{-j,-j}^{-1} (x^{-j} - \mu^{-j}).$$

Estimating $\rho_j = \mathbb{E}[X^j \mid X^{-j}, y]$ may seem at odds with the model-X philosophy, which avoids modeling the $X$-$y$ dependency. However, as in standard knockoffs, there is no need to accurately learn $\rho_j$. It suffices that $\widehat{\rho}_j$ is correct under the null, where $\rho_j$ reduces to $\nu_j$, which is simple.

Moreover, since we compare two regressors with distinct input features, approximate exchangeability requires that they remain close under the null. This motivates the use of a regularized empirical risk to enforce stability (or robustness) with respect to the inclusion of non-informative features:

$$R_n(\theta) := \frac{1}{n} \sum_{i=1}^n l(\theta^\top \chi_i, z_i) + \lambda \|\theta\|^2, \tag{2}$$

for $\lambda > 0$. In this risk we have denoted the target as $z$ rather than $X^j$, and the input as $\chi$ (resp. $X^{-j}$) rather than $(X^{-j}, y)$ (resp. $X^{-j}$) when estimating $\rho_j$ (resp. $\nu_j$). We retain this notation throughout this section, as the resulting stability statement is general and of independent interest.

Our stability result states that, the estimator trained with the additional uninformative feature $X^j$ and the estimator trained without it remain sufficiently close. In particular, their regression coefficients differ only by a small amount. While stability results for the removal or replacement of data points exist (see, e.g., Bousquet & Elisseeff (2002)), to the best of our knowledge there are no analogous results addressing the removal of uninformative *features*.

Let's denote by $\widehat{\theta}$ (resp. $\widehat{\theta^{-j}}$) the minimizer of the empirical risk (2) in $\mathbb{R}^p$ (resp. $\mathbb{R}^{p-1}$ without the $j$-th coordinate). We denote by $\widetilde{\theta}^j = (0, \widehat{\theta^{-j}})$ the restricted optimizer extended by 0 in the $j$-th coordinate.

**Theorem 4.1** (Optimization stability). *For $j \in \mathcal{H}_0$, under the assumptions stated in Appendix E.3.1, we have, with probability greater than $1 - \delta$, that*

$$\|\widetilde{\theta}^j - \widehat{\theta}\|_2 \le O_P\left(\sqrt{\frac{\log(1/\delta)}{n}}\right). \tag{3}$$

It relies on regularity assumptions on the loss, which we verify for the quadratic and cross-entropy. We also assume standard conditions on the data distribution (e.g., subgaussianity). As a consequence, $\widehat{\nu}$ and $\widehat{\rho}$ are close, as illustrated by the blue dots in Figure 1, which show the difference in

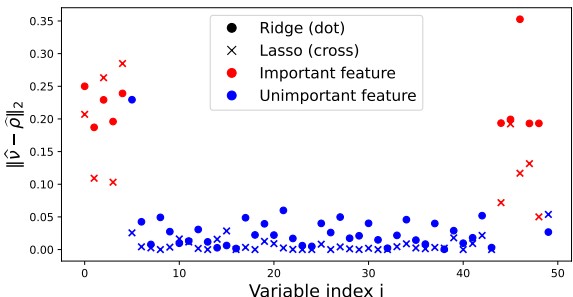

*Figure 1.* **Optimization stability.** Data are generated from $z = \chi\beta + \epsilon$, where $\beta$ is 0.25-sparse with important features grouped in blocks of 5 sampled uniformly. We set $n = 300$, $p = 50$, noise level at $\|\chi\beta\|/2$ and $\chi \sim \mathcal{N}(0, \Sigma)$ with $\Sigma_{i,j} = 0.6^{|i-j|}$. The $y$-axis shows the difference between the model coefficients with and without the $x$-axis coordinate.

the imputer's coefficients when suppressing an unimportant feature. In contrast, the red dots exhibit larger changes since the suppressed coordinate is important. Although the result applies only to a l2-regularized imputer, we observe similar behavior for Lasso in practice. It mainly reflects the desirable property that adding a noisy variable should not substantially affect the model. In Section 4.5 we provide additional arguments based on a double-robustness property.

### 4.3. Distribution control

The validity of the method relies on the fact that, under the null, the two imputers $\widehat{\rho}_j$ and $\widehat{\nu}_j$ are sufficiently close. Then, both their predictions and the residuals from which we sample are similar. Hence, we are sampling independently from two distributions that converge . Since our test statistic is based on their difference, this guarantees the required exchangeability.

We recover the Semi-knockoffs notation: $\widehat{\nu}_j$ is the regressor of $X^j$ on $X^{-j}$, and $\widehat{\rho}_j$ the one of $X^j$ on $(X^{-j}, y)$. We denote by $\hat{P}_1^j$ and $\hat{P}_2^j$ the distributions induced by sampling with $\widehat{\nu}_j$ and $\widehat{\rho}_j$, respectively, that is, the laws of

$$l(\widehat{m}(\widetilde{X}_1'^{(j)}), y) \quad \text{and} \quad l(\widehat{m}(\widetilde{X}_2'^{(j)}), y).$$

We work under the assumptions of the previous section and also assume that the model is sufficiently regular, satisfying a Lipschitz condition, together with a Lipschitz loss on its prediction argument, as in Bottou & Bousquet (2007).

**Theorem 4.2** ($\mathcal{W}_1$ control)**.** *For $j \in \mathcal{H}_0$, under Theorem 4.1 assumptions and that model $\widehat{m}$ and loss in the prediction argument are Lipschitz, we have, with probability at least $1 - \delta$, that*

$$\mathcal{W}_1(\hat{P}_1^j, \hat{P}_2^j) \leq O_P\left(\sqrt{\frac{\log(1/\delta)}{n}}\right).$$

This result connects the parameter-level stability from Theorem 4.1 to distributional stability by bounding, in Wasserstein distance, the distributions from which we sample.

### 4.4. Double robustness

Reyero-Lobo et al. (2025a) introduced the concept of *double robustness* for the CFI, which is similarly based on an estimated model $\widehat{m}$ and conditional sampler $\widehat{P}$. They showed that, for the Gaussian linear model and using the regression-based conditional sampler $\widehat{\nu}$, when both $\widehat{m}$ and $\widehat{\nu}$ are well estimated, the CFI converges to zero at a faster rate (quadratic rather than linear) under the null. Here, we generalize this perspective and, under smoothness assumptions on $\widehat{m}$, show that the decay rates associated with the two models compound, yielding this faster convergence.

**Theorem 4.3** (Double robustness)**.** *Assume that $j \in \mathcal{H}_0$ and that $t \to l(t, y)$ is $C^2$ with bounded derivatives. Assume also that $\widehat{m}$ is differentiable in the $j$-coordinate and denote by $a_n := \partial_{x^j}\widehat{m}(X^{-j}, t)$ for a $t$ between $\widetilde{X}'^j$ and $\widetilde{X}^j$. Assume that $\widehat{\nu} - \nu = O_P(b_n)$. Then,*

$$l(\widehat{m}(\widetilde{X}'), y) - l(\widehat{m}(\widetilde{X}), y) = O_P(a_n b_n).$$

As we will see in the next section, strict differentiability is not necessary in practice. For instance, with Random Forests, we still observe faster convergence. Indeed, several consistency results have highlighted their adaptability to sparse settings, showing that splits are rarely made on null features (Scornet et al., 2015; Klusowski, 2021), leading to the corresponding feature being effectively insignificant in the model. Other methods, such as Neural Networks, can be trained with an explicit penalization on the gradients to improve generalization (Zhao et al., 2022). However, as seen below, this is not necessary to achieve the faster rate.

*Remark* 4.4. This indeed generalizes the faster rates of the Gaussian linear model obtained in Reyero-Lobo et al. (2025a). Specifically, for $\widehat{m}(X) = \widehat{\beta}^\top X$, we have $\partial_{x^j}\widehat{m}(X) = \widehat{\beta}^j$, which, under the null hypothesis, is centered around 0 with variance of order $O(1/n)$. Moreover, using Gaussianity, $\nu$ is linear. This yields the desired result, since $\widehat{\nu}$ also achieves the same parametric rate. The same reasoning extends naturally to any generalized linear model.

### 4.5. Applications of double robustness to Semi-knockoffs

Note that the validity of Semi-knockoffs mainly relies on *exchangeability under the null*. This property follows from $X_1$ and $X_2$ being sampled independently from the same distribution, and from the fact that under the null $\rho(X^{-j}, y) := \mathbb{E}[X^j \mid X^{-j}, y] = \mathbb{E}[X^j \mid X^{-j}] =: \nu(X^{-j})$. Consequently, for any model $\widehat{m}$, we have

$$\text{sign}\left(l(\widehat{m}(\widetilde{X}_1), y) - l(\widehat{m}(\widetilde{X}_2), y)\right) \overset{d}{=} \epsilon^j,$$

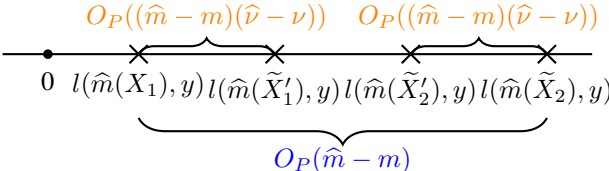

*Figure 2.* Illustration of the conjecture that $\mathbb{P}\Big( l(\widehat{m}(\widetilde{X}_1'), y) > l(\widehat{m}(\widetilde{X}_2'), y) \,\Big|\, l(\widehat{m}(\widetilde{X}_1), y) > l(\widehat{m}(\widetilde{X}_2), y) \Big) \longrightarrow 1$, thanks to the double robustness property.

where $\epsilon^j \sim \mathcal{U}(\{-1, +1\})$ is a Rademacher r.v.

We would like the same result to hold with high probability for the empirical counterparts $\widetilde{X}_1'$ and $\widetilde{X}_2'$ based on $\widehat{\nu}$ and $\widehat{\rho}$. We conjecture that this is the case, based on a double robustness argument. Indeed, we expect that the sign of the empirical difference matches the sign of the theoretical one, yielding the desired independent Rademacher variable. Formally, if we arbitrarily assume that the difference is positive, i.e.,

$$l(\widehat{m}(\widetilde{X}_1), y) - l(\widehat{m}(\widetilde{X}_2), y) > 0, \qquad (4)$$

we expect the same sign for the empirical counterparts. This expectation is justified because the difference in (4) is of the order of the insensitivity of the model $\widehat{m}$ with respect to the $j$-th coordinate, while the differences

$$l(\widehat{m}(\widetilde{X}_1), y) - l(\widehat{m}(\widetilde{X}_1'), y) \text{ and } l(\widehat{m}(\widetilde{X}_2), y) - l(\widehat{m}(\widetilde{X}_2'), y)$$

decay according to both $\widehat{m}$ and $\widehat{\nu}$ (or $\widehat{\rho}$), then faster, as formalized in Theorem 4.3 and illustrated in Figure 2.

Figure 3 presents the exchangeability idea through double robustness in practice. The same experiment using other learners and studying the effect of independence among training samples is discussed in Appendix F.2. For a null feature ($X^0$), we plot in blue the distribution capturing only the decay resulting from the sensitivity of $\widehat{m}$ to coordinate $j = 0$, obtained by comparing the performance on two independent samples. We incorporate in orange the decay from both $\widehat{m}$ and the imputers $\widehat{\rho}$ and $\widehat{\nu}$ by comparing the theoretical and empirical residuals from the same sample. Even when employing complex learners for $\widehat{m}$ (Random Forests, Gradient Boosting, and Neural Networks), we observe that the distribution enjoying double robustness (orange) is substantially more concentrated around zero, as established by Theorem 4.3. Moreover, in practice, our *knockoff statistic* $W_{\text{SKO}}^j$ is always based on the distribution in blue corresponding to the fully estimated models, which remains symmetric and centered at zero across all three complex learners, thereby preserving the desired exchangeability property.

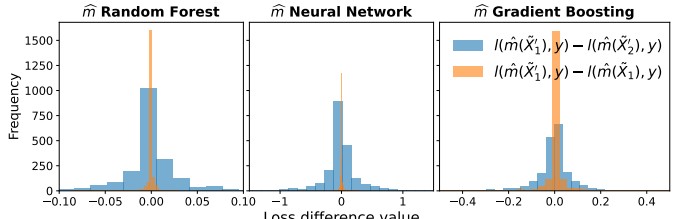

*Figure 3.* **Empirical evidence for Double Robustness:** Distribution of the Semi-knockoff statistic, i.e., the difference in loss evaluated at two independently sampled estimated residuals (blue: $l(\widehat{m}(\widetilde{X}_1'), y) - l(\widehat{m}(\widetilde{X}_2'), y)$), and distribution of the difference between the theoretical and estimated imputer (orange: $l(\widehat{m}(\widetilde{X}_1'), y) - l(\widehat{m}(\widetilde{X}_1), y)$) for a null coordinate ($j = 0$). The data is sampled from $y = 0.8X^1 + 0.6X^2 + 0.4X^3 + 0.2X^4 + \sin(X^1) + \epsilon$, with $\epsilon \sim \mathcal{N}(0, 0.5)$. We use $n = 2000$ samples with $X \sim \mathcal{N}(0, \Sigma)$, where $\Sigma^{i,j} = 0.5^{|i-j|}$. $\widehat{\nu}$ is a linear model.

## 5. Experiments

We compare Variable Importance Measures (VIM), such as CFI (Strobl et al., 2008), with valid tests given by the square-root additive correction `SCPI(1)_sqrt` and Wilcoxon test `SCPI(1)_Wcx` from (Reyero-Lobo et al., 2025a). We also consider the SAGE value function (Covert et al., 2020) with the correction term `SCPI(100)_sqrt`, and LOCO (Lei et al., 2018) with extra data splitting `LOCO-W` (Williamson et al., 2023), the additive correction `LOCO_sqrt` (Verdinelli & Wasserman, 2024), or the Wilcoxon test. From Variable Selection, we include dCRT (Liu et al., 2022) and HRT (Tansey et al., 2022). A summary of the methods, references and guarantees in given in Table 2 in the Supplementary Materials.

We present only the Semi-knockoffs with the Wilcoxon test, `SKO_Wcx`, since it is typically more powerful than the sign test. Moreover, it is possible to *Rao–Blackwellize* the procedure by performing multiple permutations per sample instead of only one; such derandomization strategies have proven beneficial in sequential randomized tests (Shaer et al., 2023). In our experiments, we already observe a clear gain when applying 5 permutations (`SKO_Wcx_p5`).

We report power, type-I error control, and AUC, and provide time analyses together with additional results for other models and settings in Appendix F. FDR control results are presented in Appendix F.5, leading to similar conclusions regarding the power gain obtained by avoiding the train–test split, as well as an additional gain from using the knockoff threshold instead of applying a BH correction to $p$-values.

All experiments are repeated 50 times with $n = 300$ and $p = 50$. Similar conclusions were obtained for $n = 100, 200,$ and $400$, with a particularly clear power gain for smaller sample sizes due to avoiding the train–test split. The code is available at

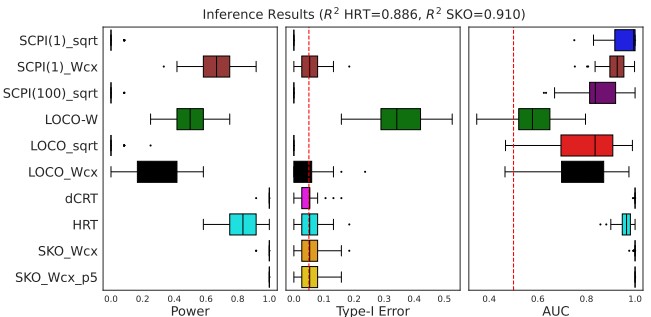

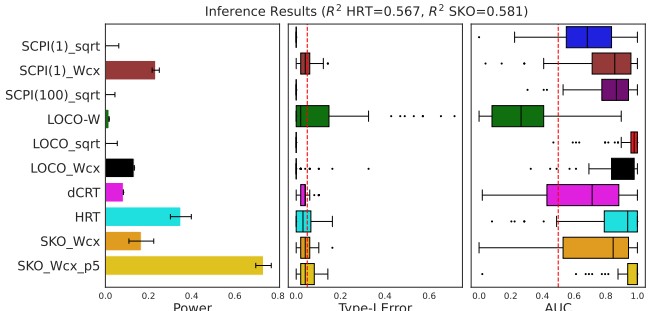

*Figure 4.* **Type-I error with adjacent support.** Let $X \sim \mathcal{N}(0, \Sigma)$ with $\Sigma^{ij} = 0.6^{|i-j|}$, and $y = \beta^\top X + \epsilon$, where the first $0.25p$ coordinates of $\beta$ lie in $[1, 2]$ and the remaining are zero, with $\epsilon \sim \mathcal{N}(0, 1)$. The black-box pretrained model $\widehat{m}$ is a gradient boosting, achieving $R^2 = 0.886$ with a train-test split and $R^2 = 0.91$ without split.

*Figure 5.* **Type-I error with masked correlation.** Let $X \sim \mathcal{N}(0, \Sigma)$ with $\Sigma_{ij} = 0.6^{|i-j|}$. A unique relevant coordinate $l$ is sampled and $y = X_l + 0.5\,\epsilon_1$, where $\epsilon_1 \sim \mathcal{N}(0, 1)$. A correlated null variable is created as $X_{l-1} = X_l + 0.5\,\epsilon_2$, with $\epsilon_2 \sim \mathcal{N}(0, 1)$. The black-box pretrained model $\widehat{m}$ is a neural network, achieving $R^2 = 0.567$ with a train-test split and $R^2 = 0.581$ without split.

https://github.com/AngelReyero/loss_based_KO.

## 5.1. Simulated data

In Figure 4 we study a linear setting with *adjacent support*. We first observe that VIMs are too restrictive to make discoveries: they typically rely on asymptotic guarantees, which leads some methods (e.g. LOCO-W) to fail to control the type-I error, and their reliance on decay rates results in low power. In contrast, variable selection methods enjoy finite-sample control and are substantially more powerful. Finally, we note a clear advantage in avoiding sample splitting, as the HRT is noticeably less powerful than methods that do not split the sample. In Figure 5 we study a *masked correlation* setting in which an important feature is strongly correlated with a null feature. As before, we observe that the corrections used in VIMs are too restrictive, highlighting the need for valid testing procedures that can still accommodate arbitrary models. In contrast, the dCRT exhibits low power: its two-step feature-splitting strategy, used to speed up computation, prevents it from capturing all relevant interactions, and thus the first distillation generally remove most of the signal from the important feature. Finally, we observe a substantial gain from derandomization of the Semi-knockoffs: using only five permutations already yields the most powerful procedure.

## 5.2. Real data

We also compare the methods on real data to evaluate their discovery capability. For this, we use the Wisconsin Breast Cancer dataset (Mangasarian et al., 1995); see Appendix F.6 for more details on the data and experiments. Since the true set of important features is unknown, we add an extra null feature correlated at $0.6$ with the original features. This allows us to estimate the type-I error in Figure 6 using this artificial feature, indicated on the right of each plot. We

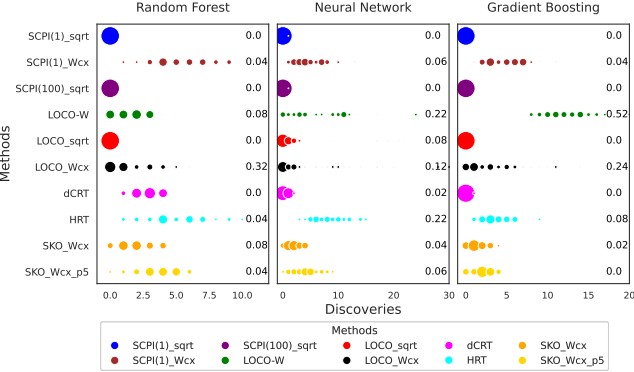

*Figure 6.* **Discoveries on real data across ML models.** The type-I error (shown to the right of each plot) is estimated using an artificial null feature correlated at $0.6$ with the original features. Semi-knockoffs make discoveries while controlling the error.

present results using Random Forests, Neural Networks, and Gradient Boosting. Similar to the simulated data, the proposed method can make discoveries while controlling the error; derandomization slightly increases power. Importantly, it exhibits stable power across models and consistent numbers of discoveries, unlike other methods whose results vary significantly with the random seed.

## 6. Discussion

We have presented *Semi-knockoffs*, a framework for CIT that accommodates any pretrained model and avoids train-test splitting. This is achieved by constructing a test statistic that is symmetric under the null through the creation of two populations: one that incorporates information from the response and one that does not. As a result, the dependence between the model and the distribution under test does not need to be broken; the model is used only as an importance score, similarly to how lasso coefficients are used. This allows us to leverage the powerful knockoff thresholding procedure

without requiring the stringent structural constraints needed to construct knockoff variables and statistics. Moreover, it provides both type-I error and FDR control.

One limitation may come from using a simple model for the imputer incorporating the response. While this remains valid, it may fail to capture richer relationships and break the symmetry with the other imputer under the alternative. Future work will investigate how such imputations and other ways of exploiting information in the response can be combined to further increase statistical power, and whether more complex imputation models can still control the error in practice via double-robustness guarantees similar to those established in this paper. Importantly, note that in the context of missing-data imputation, sampling strategies that condition on both covariates and the response have recently been explored and they have shown substantial improvements (D'Agostino McGowan et al., 2024).

## Acknowledgements

This work benefited from state aid managed by the Agence Nationale de la Recherche ANR-23-CE23-0016 and the H2020 Research Infrastructures Grant EBRAIN-Health 101058516.

## Impact Statement

This paper presents work whose goal is to advance the field of Machine Learning. There are many potential societal consequences of our work, none which we feel must be specifically highlighted here.

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

# A. Notation Glossary

| Symbol | Description |
| --- | --- |
| $X \in \mathbb{R}^p$ | Input |
| $X^j \in \mathbb{R}$ | $j$-th input covariate |
| $X^{-j} \in \mathbb{R}^{p-1}$ | Input with the $j$-th covariate excluded |
| $y \in \mathbb{R}$ | Output |
| $x_i$ | $i$-th individual |
| $x_i^{(j)}$ | $i$-th individual with permuted $j$-th covariate |
| $m(X)$ (resp. $m_{-j}(X^{-j})$) | $\mathbb{E}\left[y \mid X\right]$ (resp. $\mathbb{E}\left[y \mid X^{-j}\right]$) |
| $\widehat{m}$ (resp. $\widehat{m}_{-j}$) | Estimation of $m$ (resp. of $m_{-j}$) |
| $\nu_j(X^{-j})$ | $\mathbb{E}\left[X^j \mid X^{-j}\right]$ |
| $\widehat{\nu}_j$ | Estimation of $\nu_j$ |
| $\rho_j(X^{-j}, y)$ | $\mathbb{E}\left[X^j \mid X^{-j}, y\right]$ |
| $\widehat{\rho}_j$ | Estimation of $\rho_j$ |
| $\widetilde{X}_1^{(j)}$ | Semi-knockoff variable drawn using $\nu_j$ |
| $\widetilde{X}_2^{(j)}$ | Semi-knockoff variable drawn using $\rho_j$ |
| $\widetilde{X}_1^{'(j)}$ | Semi-knockoff variable drawn using $\widehat{\nu}_j$ |
| $\widetilde{X}_2^{'(j)}$ | Semi-knockoff variable drawn using $\widehat{\rho}_j$ |
| $\hat{P}_1^j$ | Distribution of $\widetilde{X}_1^{'(j)}$ |
| $\hat{P}_2^j$ | Distribution of $\widetilde{X}_2^{'(j)}$ |
| $\ell$ | Loss function |
| $\mathcal{W}_1$ | 1-Wasserstein distance |
| $\mathcal{L}$ | Likelihood |
| $T_q$ | Knockoff threshold |
| $W_{\text{SKO}}^j$ (resp. $\hat{W}_{\text{SKO}}^j$) | Oracle (resp. Practical) Semi-knockoff statistic |
| $S_{\text{SKO}}$ (resp. $\hat{S}_{\text{SKO}}$) | Oracle (resp. Practical) Semi-knockoff selected features |
| $R_n$ | Regularised empirical risk |
| $O_P(a_n)$ | Bounded in probability at a rate of $a_n$. |

The superscript $(j)$ is omitted to avoid index overload when $j$ can be inferred from the context.

# B. Explicit definitions

For the sake of space, in this section we provide explicit definitions of the procedures that were introduced in Section 2.

## B.1. Leave One Covariate Out (LOCO)

LOCO consists of assessing the importance of a feature by studying the performance drop when the model is trained without it ($\widehat{m}_{-j}$).

**Definition B.1** (LOCO). Given $j$, a loss $\ell$, a regressor $\widehat{m}$ of $y$ given $X$, a regressor $\widehat{m}_{-j}$ of $y$ given $X^{-j}$ and a test set $(X_i, y_i)_{i=1,\ldots n_{\text{test}}}$, LOCO is defined as

$$\widehat{\psi}_{\text{LOCO}}^j = \frac{1}{n_{\text{test}}} \sum_{i=1}^{n_{\text{test}}} \ell\left(\widehat{m}_{-j}(x_i^{-j}), y_i\right) - \ell\left(\widehat{m}(x_i), y_i\right).$$

## B.2. Conditional Feature Importance (CFI)

CFI consists of reusing the same model while restricting the information from the feature by breaking its link with the output, while preserving its dependence with the rest of the input to avoid extrapolation.

**Definition B.2** (CFI). Given $j$, a loss $\ell$, a regressor $\widehat{m}$ of $y$ given $X$ and a test set $(X_i, y_i)_{i=1,\ldots n_{\text{test}}}$, CPI is defined as

$$\widehat{\psi}_{\text{CPI}}^j = \frac{1}{n_{\text{test}}} \sum_{i=1}^{n_{\text{test}}} \ell\left(\widehat{m}(\widetilde{x}_i'^{(j)}), y_i\right) - \ell\left(\widehat{m}(x_i), y_i\right),$$

where the $j$-th coordinate is *conditionally* permuted.

### B.3. Knockoff definition

Knockoffs (Candès et al., 2018) provides a framework for studying conditional independence testing with FDR control by creating synthetic features that appear feasible but break the link with the output, and then comparing the performance between the knockoff features and the original ones. More formally, it relies on *knockoff variables* $\widetilde{X}$ (Appendix B.3.1), which do not preserve the relationship with $y$ but follow the distribution of $X$, a *knockoff statistic* $W$ (Appendix B.3.2) that assigns an importance to the feature, and the *knockoff threshold* $T_q$ (1).

#### B.3.1. KNOCKOFF VARIABLES

**Definition B.3** (Model-X knockoffs). For a random variable $X$, it is a new random variable $\widetilde{X}$ constructed satisfying the following two properties:

1. For any subset $s \subset \{1, \ldots, p\}$, the original and the knockoff variables are exchangeable, i.e. $(X, \widetilde{X})_{\text{swap}(s)} \overset{d}{=} (X, \widetilde{X})$.

2. $\widetilde{X} \perp\!\!\!\perp y | X$.

We observe that the second property can be obtained by constructing the knockoffs without using the output. Moreover, these imitation variables must satisfy the exchangeability condition that is stronger than merely sampling from the conditional distribution, as it must hold over the joint distribution rather than feature-wise. Consequently, computing knockoffs in practice requires a sequential procedure and cannot be parallelized, in contrast to the CRT. While many methods have been proposed for generating knockoff variables in practice (Sesia et al., 2019; Romano et al., 2020; Bates et al., 2021), and recent work has established asymptotic FDR control (Fan et al., 2025), concerns have been raised regarding their finite-sample validity (Blain et al., 2025).

#### B.3.2. KNOCKOFF STATISTICS

**Definition B.4** (Feature statistics from (Candès et al., 2018)). It is a vector $\mathbf{W} = (W_1, \ldots, W_p) \in \mathbb{R}^p$ where each coordinate $W_j$ satisfies

1. It is a function of the input $X$, the knockoff $\widetilde{X}$ and the output $y$: $W_j = w_j\left(\left[X, \widetilde{X}\right], y\right)$.

2. It satisfies the flip-sign property: for $s \subset [p]$,

$$w_j\left(\left[X, \widetilde{X}\right]_{\text{swap}(s)}, y\right) = \begin{cases} w_j\left(\left[X, \widetilde{X}\right], y\right) & \text{if } j \not\in s, \\ -w_j\left(\left[X, \widetilde{X}\right], y\right) & \text{if } j \in s. \end{cases}$$

This statistic provides a hint of how important the feature is. The most intuitive example is given by the Lasso Coefficient Difference (LCD), which, to assign importance, compares the coefficients of a Lasso regression of $y$ on both $(X, \widetilde{X})$, so that $W_{\text{LCD}}^j := |\hat{\beta}^j| - |\hat{\beta}^{j+p}|$. Then, if the coordinate is important, there is more information in the coefficient for the original feature than for its knockoff counterpart.

## C. Algorithms

We first present the theoretical Semi-knockoffs, which provide valid type-I error control through nonparametric paired tests (Wilcoxon or Sign test), and then describe the procedure for controlling FDR using the knockoff threshold. We note that, in practice, the procedure is identical, relying on the computation of the regressors $\widehat{\nu}$ and $\widehat{\rho}$ (Algorithm 4).

## C.1. Type-I error algorithm (Semi-KO Sign test/Wilcoxon)

---

**Algorithm 2** Oracle Semi-knockoffs Sign-Test/Wilcoxon (Semi-KO ST/Wilcox)

---

**Input:** Model $\widehat{m}$, data $\{(X_i, y_i)\}_i^n$, significance level $\alpha$, a feature $j$, conditional expectations $\nu_j$ and $\rho_j$

Compute theoretical residuals $\epsilon_{j,1} = X^j - \nu_j(X^{-j})$

Compute theoretical residuals $\epsilon_{j,2} = X^j - \rho_j(X^{-j}, y)$

Sample $\pi_{j,1}$ and $\pi_{j,2}$ permutations of $\{1, \ldots, n\}$

Construct

$$\widetilde{X}_{1,i}^{(j)} = \nu_j(X_i^{-j}) + \epsilon_{j,1,\pi_{j,1}(i)}$$
$$\widetilde{X}_{2,i}^{(j)} = \rho_j(X_i^{-j}, y_i) + \epsilon_{j,2,\pi_{j,2}(i)}$$

Compute nonparametric paired test (Sign-test/Wilcoxon) between

$$\left\{ l\big(\widehat{m}(\widetilde{X}_{1,i}^{(j)}), y_i\big) \right\} \text{ and } \left\{ l\big(\widehat{m}(\widetilde{X}_{2,i}^{(j)}), y_i\big) \right\}$$

**Return** the computed p-value

---

## C.2. Oracle Semi-knockoffs

---

**Algorithm 3** Oracle Semi-knockoffs (Semi-KO)

---

**Input:** Model $\widehat{m}$, data $\{(X_i, y_i)\}_i^n$, significance level $q$, conditional expectations $\nu_j$ and $\rho_j$ for every feature $j$

**For** $j = 1, \ldots, p$

Compute theoretical residuals $\epsilon_{j,1} = X^j - \nu_j(X^{-j})$

Compute theoretical residuals $\epsilon_{j,2} = X^j - \rho_j(X^{-j}, y)$

Sample $\pi_{j,1}$ and $\pi_{j,2}$ permutations of $\{1, \ldots, n\}$

Construct

$$\widetilde{X}_{1,i}^{(j)} = \nu_j(X_i^{-j}) + \epsilon_{j,1,\pi_{j,1}(i)}$$
$$\widetilde{X}_{2,i}^{(j)} = \rho_j(X_i^{-j}, y_i) + \epsilon_{j,2,\pi_{j,2}(i)}$$

Compute the statistic

$$W_{\text{SKO}}^j = \frac{1}{n} \sum_{i=1}^n \Big( l\big(\widehat{m}(\widetilde{X}_{1,i}^{(j)}), y_i\big) - l\big(\widehat{m}(\widetilde{X}_{2,i}^{(j)}), y_i\big) \Big)$$

Compute knockoff threshold $T_q$ as in (1)

**Return**

$$S_{\text{SKO}} = \big\{ j : W_{\text{SKO}}^j \geq T_q \big\}.$$

---

## C.3. Semi-knockoffs

---

**Algorithm 4** Semi-knockoffs (SKO)

---

**Input:** Model $\widehat{m}$, data $\{(X_i, y_i)\}_i^n$, significance level $q$

**For** $j = 1, \ldots, p$

   (**Regression 1**) Fit $\widehat{\nu}_j \approx \mathbb{E}[X^j \mid X^{-j}]$

   (**Regression 2**) Fit $\widehat{\rho}_j \approx \mathbb{E}[X^j \mid X^{-j}, y]$

   Compute residuals $\widehat{\epsilon}_{j,1} = X^j - \widehat{\nu}_j(X^{-j})$

   Compute residuals $\widehat{\epsilon}_{j,2} = X^j - \widehat{\rho}_j(X^{-j}, y)$

   Sample $\pi_{j,1}$ and $\pi_{j,2}$ permutations of $\{1, \ldots, n\}$

   Construct

$$\widetilde{X}'^{(j)}_{1,i} = \widehat{\nu}_j(X_i^{-j}) + \widehat{\epsilon}_{j,1,\pi_{j,1}(i)}$$
$$\widetilde{X}'^{(j)}_{2,i} = \widehat{\rho}_j(X_i^{-j}, y_i) + \widehat{\epsilon}_{j,2,\pi_{j,2}(i)}$$

   Compute the statistic

$$\hat{W}^j_{\text{SKO}} = \frac{1}{n} \sum_{i=1}^n \left( l\big(\widehat{m}(\widetilde{X}'^{(j)}_{1,i}), y_i\big) - l\big(\widehat{m}(\widetilde{X}'^{(j)}_{2,i}), y_i\big) \right)$$

   Compute knockoff threshold $T_q$ as in (1)

   **Return**

$$\widehat{S}_{\text{SKO}} = \big\{ j : \hat{W}^j_{\text{SKO}} \geq T_q \big\}.$$

---

# D. Mathematical tools

In this section, we present some mathematical tools and results from high-dimensional probability that were used in our analysis. They are mainly obtained from Vershynin (2018). The first definitions concern the sub-Gaussian and sub-exponential norms:

**Definition D.1** (Subgaussian norm). $\|X\|_{\psi_2} = \inf\{K > 0, \mathbb{E}\left[\exp X^2/K^2\right] \leq 2\}$.

**Definition D.2** (Subexponential norm). $\|X\|_{\psi_1} = \inf\{K > 0, \mathbb{E}\left[\exp|X|/K\right] \leq 2\}$.

Also, Lemma 2.8.6 from Vershynin (2018) gives that the product of subgaussians is subexponential.

**Lemma D.3** (Subgaussian × subgaussian = subexponential). *If $X$ and $Y$ are subgaussian, then $XY$ is subexponential, and* $\|XY\|_{\psi_1} \leq \|X\|_{\psi_2}\|Y\|_{\psi_2}$.

Similarly to sub-Gaussian random variables, there is an analogous Bernstein-type result for sub-exponential variables (Theorem 2.9.1 in Vershynin (2018)):

**Theorem D.4** (Subexponential Bernstein). *Let $X_1, \ldots, X_n$ independent, mean zero subexponential random variables. Then, for every $t \geq 0$,*

$$\mathbb{P}\left( \left| \sum_{i=1}^n X_i \right| \geq t \right) \leq 2\exp\left( -c\min\left( \frac{t^2}{\sum_{i=1}^n \|X_i\|_{\psi_1}^2}, \frac{t}{\max_i \|X_i\|_{\psi_1}} \right) \right)$$

*where $c > 0$ is an absolute constant.*

# E. Proofs

## E.1. Semi-knockoffs type-I error control: Theorem 3.3

*Proof.* Since the algorithm relies on a nonparametric test between two populations, the proof reduces to showing that under the null hypothesis, i.e. $X^j \perp\!\!\!\perp y \mid X^{-j}$, both sample distributions come from the same population. This follows because conditional independence implies, in particular, independence of the first-order moment. Therefore, both $\rho$ and $\nu$ coincide:

$$\rho_j(X^{-j}, y) = \mathbb{E}[X^j \mid X^{-j}, y] = \mathbb{E}[X^j \mid X^{-j}] = \nu_j(X^{-j}).$$

In particular, we have that

$$l(\widehat{m}(\widetilde{X}_1^{(j)}), y) \stackrel{\mathrm{d}}{=} l(\widehat{m}(\widetilde{X}_2^{(j)}), y).$$

$\square$

### E.2. Semi-knockoffs FDR control

*Proof.* To control the FDR, the only thing that we need to prove is the exchangeability. Once this is proved, we can conclude using the same proofs as in Theorem 1 and 2 from (Barber & Candès, 2015). For this proof, we include $2p$ i.i.d uniform random variables $U_1, \dots U_p, U_1', \dots U_p' \sim \mathcal{U}\{1, \dots, n\}$ where $U_j$ (resp. $U_j'$) represents the choice of the permutation that is done for the $\widetilde{X}_1^{(j)}$ (resp. for the $\widetilde{X}_2^{(j)}$). We observe that this is indeed correct since each sampling for each residual is done independently. Then, we denote by $(X_{U_j}, y_{U_j})$ the sample used for the residual of $\widetilde{X}_1^{(j)}$ and similarly by $(X_{U_j'}, y_{U_j'})$ for $\widetilde{X}_2^{(j)}$.

Also, we note that under the null hypothesis, since $X^j \perp\!\!\!\perp y \mid X^{-j}$, we have that $\rho_j(X^{-j}, y) := \mathbb{E}\left[ X^j \mid X^{-j}, y \right] = \mathbb{E}\left[ X^j \mid X^{-j} \right] =: \nu_j(X^{-j})$. Thus, we have that for $j \in \mathcal{H}_0$:

$$
\begin{aligned}
\widetilde{X}_1^{(j)} &:= \nu_j(X^{-j}) + \left( X_{U_j}^j - \nu_j(X_{U_j}^{-j}) \right) \\
&= \rho_j(X^{-j}, y) + \left( X_{U_j}^j - \rho_j(X_{U_j}^{-j}, y_{U_j}) \right) \\
&\stackrel{\mathrm{d}}{=} \rho_j(X^{-j}, y) + \left( X_{U_j'}^j - \rho_j(X_{U_j'}^{-j}, y_{U_j'}) \right) \\
&=: \widetilde{X}_2^{(j)}.
\end{aligned}
$$

Finally, we note that the statistic vector provided by the Semi-knockoffs, $\mathbf{W}_{\mathrm{SKO}}$, has the same distribution if we change the sign of any coordinate $j \in \mathcal{H}_0$:

$$
\begin{aligned}
\mathbf{W}_{\mathrm{SKO}} &:= \left( l\left( \widehat{m}\left( \widetilde{X}_1^{(1)} \right), y \right) - l\left( \widehat{m}\left( \widetilde{X}_2^{(1)} \right), y \right), \dots, l\left( \widehat{m}\left( \widetilde{X}_1^{(j)} \right), y \right) - l\left( \widehat{m}\left( \widetilde{X}_2^{(j)} \right), y \right), \right. \\
&\qquad \left. \dots, l\left( \widehat{m}\left( \widetilde{X}_1^{(p)} \right), y \right) - l\left( \widehat{m}\left( \widetilde{X}_2^{(p)} \right), y \right) \right) \\
&\stackrel{\mathrm{d}}{=} \left( l\left( \widehat{m}\left( \widetilde{X}_1^{(1)} \right), y \right) - l\left( \widehat{m}\left( \widetilde{X}_2^{(1)} \right), y \right), \dots, l\left( \widehat{m}\left( \widetilde{X}_2^{(j)} \right), y \right) - l\left( \widehat{m}\left( \widetilde{X}_1^{(j)} \right), y \right), \right. \\
&\qquad \left. \dots, l\left( \widehat{m}\left( \widetilde{X}_1^{(p)} \right), y \right) - l\left( \widehat{m}\left( \widetilde{X}_2^{(p)} \right), y \right) \right).
\end{aligned}
$$

We observe that this can be done for any $j \in \mathcal{H}_0$ independently, since the permutations are independent. Thus, by applying the previous argument to all indices in this null set, we obtain exchangeability.

$\square$

### E.3. Null feature stability of the ERM

In this section, we provide a proof for Theorem 4.1. To do so, we begin by presenting the necessary assumptions in Section E.3.1, followed by the proof in Section E.3.2.

#### E.3.1. ASSUMPTIONS FOR THE STABILITY RESULT

Our first assumption concerns the regularity of the score function with respect to its first argument.

**Assumption E.1** (Lipschitz score)**.** The score is $L$-Lipschitz with respect to the predictions:

$$|s(u_1, y) - s(u_2, y)| \le L \|u_1 - u_2\|.$$

We easily observe that this is achieved for instance for the quadratic loss and also for the cross-entropy loss by bounding the probability of belonging to a class by an $0 < a < b < 1$.

We assume that the input matrix as well as the score are subgaussian, i.e., their subgaussian norms (see Definition D.1) are finite.

**Assumption E.2** (Design). $\chi$ is subgaussian with covariance $\Sigma$ with bounded operator norm.

**Assumption E.3** (Score). The score in $\theta^\star$ given by $s_i^\star := s_i(\theta^\star) = \partial_u l(\theta^{\star\top}\chi_i, z_i)$ is subgaussian.

This is, for example, satisfied when using the quadratic loss together with subgaussian input features and output.

Finally, we assume that the population minimizer achieves a zero expected derivative of the loss, as it is the minimizer of the population risk.

**Assumption E.4** (Population minimizer). $\mathbb{E}\left[\partial_u l(\mu(\chi), z) \mid \chi\right] = 0$ for $\mu(\chi) = \mathbb{E}\left[z \mid \chi\right]$.

In Lemma E.5 we verify this assumption for the quadratic loss and the cross-entropy loss. More generally, this property holds for many other losses when choosing the corresponding population minimizer, since such minimizers are typically defined to satisfy this condition whenever the loss is differentiable (and even in the non-differentiable case, an analogous statement can be made using subgradients; however, we restrict our presentation to the simpler differentiable setting).

**Lemma E.5.** *Denote by $\mu(\chi) := \mathbb{E}\left[z \mid \chi\right]$. Then, $\mathbb{E}\left[\partial_u l(\mu(\chi), z) \mid \chi\right] = 0$ when $l$ is the quadratic loss or the cross entropy.*

Overall, this could be generalized to other losses when picking other population minimizers, since in general they are constructed to exactly fulfil this condition when differentiable (even in this case it can be generalized via the subgradient, but we again prefer to keep the exposition simple).

*Proof.* We start with the quadratic loss, for which $\partial_u l(\mu(\chi), z) = 2(\mu(\chi) - z)$. Therefore, we observe easily by definition of $\mu$ that

$$\mathbb{E}\left[\partial_u l(\mu(\chi), z) \mid \chi\right] = \mathbb{E}\left[2(\mu(\chi) - z) \mid \chi\right] = 2(\mu(\chi) - \mathbb{E}\left[z \mid \chi\right]) = 0.$$

Similarly, for the cross-entropy, which is given by $l(u, z) = z\log(u) - (1 - z)\log(1 - u)$, we observe that $\partial_u l(\mu(\chi), z) = \frac{z}{\mu(\chi)} - \frac{1-z}{1-\mu(\chi)}$. Thus, we conclude that

$$\mathbb{E}\left[\partial_u l(\mu(\chi), z) \mid \chi\right] = \mathbb{E}\left[\frac{z}{\mu(\chi)} - \frac{1-z}{1-\mu(\chi)} \mid \chi\right] = \frac{\mathbb{E}\left[z \mid \chi\right]}{\mu(\chi)} - \frac{1 - \mathbb{E}\left[z \mid \chi\right]}{1 - \mu(\chi)} = 0.$$

$\square$

### E.3.2. PROOF OF OPTIMIZATION STABILITY: THEOREM 4.1

**Theorem E.6** (Optimization stability). *For $j \in \mathcal{H}_0$, under the assumptions stated in Appendix E.3.1, we have, with probability greater than $1 - \delta$, that*

$$\|\widetilde{\theta} - \widehat{\theta}\|_2 \leq O_P\left(\sqrt{\frac{\log(1/\delta)}{n}}\right). \tag{5}$$

*Proof.* In this proof we denote by $K_\chi := \|\chi\|_{\psi_2}$ and $K_s := \|s^\star\|_{\psi_2}$ the subgaussian norm (see Definition D.1) of the design matrix and the score at $\theta^\star$ respectively.

We start by noting that $R_n$ is $\lambda$-strongly convex, with $\lambda > 0$. In particular, we have that

$$R_n(\widehat{\theta}) \geq R_n(\widetilde{\theta}) + \nabla R_n(\widetilde{\theta})^\top(\widehat{\theta} - \widetilde{\theta}) + \frac{\lambda}{2}\|\widehat{\theta} - \widetilde{\theta}\|^2.$$

Since $\widehat{\theta}$ is the minimizer, we have that $R_n(\widehat{\theta}) \leq R_n(\widetilde{\theta})$. Thus, we have that

$$\frac{\lambda}{2}\|\widehat{\theta} - \widetilde{\theta}\|^2 \leq \nabla R_n(\widetilde{\theta})^\top(\widehat{\theta} - \widetilde{\theta}) \leq \|\nabla R_n(\widetilde{\theta})\|\|(\widehat{\theta} - \widetilde{\theta})\|,$$

where in the last inequality we used Cauchy-Schwarz. This gives us

$$\|\widehat{\theta} - \widetilde{\theta}\| \le \frac{2}{\lambda}\|\nabla R_n(\widetilde{\theta})\|.$$

Now, note that since by definition

$$\widehat{\theta^{-j}} = \mathrm{argmin}_{\theta \in \mathbb{R}^{p-1}} R_n(\theta),$$

then $\nabla_l R_n(\widetilde{\theta}) = 0$ for $l \ne j$. Thus, we simplify

$$\|\nabla R_n(\widetilde{\theta})\| = |\nabla_j R_n(\widetilde{\theta})| = \left| \frac{\partial}{\partial \theta_j} \frac{1}{n} \sum_{i=1}^n l(\theta^\top \chi_i, z_i) + \lambda \|\theta\|^2 \right|_{\widetilde{\theta}} = \left| \frac{1}{n} \sum_{i=1}^n \chi_{ij} \partial_u l(\widetilde{\theta}^\top \chi_i, z_i) + 2\lambda \widetilde{\theta}^j \right|,$$

where $\partial_u l(\widetilde{\theta}^\top \chi_i, z_i)$ is the derivate of the loss with respect to the first argument. Note that $\widetilde{\theta}^j = 0$ by definition. Thus, combining both, we remark that

$$\|\widehat{\theta} - \widetilde{\theta}\| \le \frac{2}{\lambda} \left| \frac{1}{n} \sum_{i=1}^n \chi_{ij} \partial_u l(\widehat{\theta^{-j}}^\top \chi_i^{-j}, z_i) \right|. \tag{6}$$

Note that this last term is the score evaluated at $\widehat{\theta^{-j}}$ since $s_i(\theta) = \partial_u l(\theta^\top \chi_i, z_i)$.

In the sequel we will bound the right-hand-side in probability. To do so, we decompose this score into (i) the score evaluated at the population parameter $\theta^\star$ and (ii) the estimation error:

$$\frac{1}{n} \sum_{i=1}^n \chi_{ij} \partial_u l(\widehat{\theta^{-j}}^\top \chi_i^{-j}, z_i) = \frac{1}{n} \sum_{i=1}^n \chi_{ij} \left( \partial_u l(\theta^{\star-j\top} \chi_i^{-j}, z_i) + \left( \partial_u l(\widehat{\theta^{-j}}^\top \chi_i^{-j}, z_i) - \partial_u l(\theta^{\star-j\top} \chi_i^{-j}, z_i) \right) \right). \tag{7}$$

The first term is a mean zero stochastic term. Indeed, note that using Assumption E.4, we exactly have that

$$\mathbb{E}\left[ \chi_i^j \partial_u l(\theta^{\star-j\top} \chi_i^{-j}, z_i) \right] = \mathbb{E}\left[ \chi_i^j \mathbb{E}\left[ \partial_u l(\theta^{\star-j\top} \chi_i^{-j}, z_i) \mid \chi \right] \right] = 0.$$

The second term in (7) is a bit more complex since it is an estimation error term and there is the dependence to take into account between the $\chi_i^j$ and the estimate $\widehat{\theta^{-j}}$. To deal with this term we start by using that the score is $L$-Lipschitz with respect to the model argument using Assumption E.1:

$$\frac{1}{n} \sum_{i=1}^n \chi_i^j \left( \partial_u l(\widehat{\theta^{-j}}^\top \chi_i^{-j}, z_i) - \partial_u l(\theta^{\star-j\top} \chi_i^{-j}, z_i) \right) \le \frac{L}{n} \sum_{i=1}^n |\chi_{ij}| \left| \widehat{\theta^{-j}}^\top \chi_i^{-j} - \theta^{\star-j\top} \chi_i^{-j} \right|$$

$$= L \left( \frac{1}{n} \chi^{-j\top} \chi^j \right)^\top (\widehat{\theta^{-j}} - \theta^{\star-j})$$

$$\tag{8}$$

$$\le L \left\| \frac{1}{n} \chi^{-j\top} \chi^j \right\| \left\| \widehat{\theta^{-j}} - \theta^{\star-j} \right\| := L \|M_j\| \|\Delta\|. \tag{9}$$

The $M_j$ term which is an estimate of the population covariance. Later we will provide a bound in probability of this quantity using Assumption E.2 since the population covariance is bounded. For the $\Delta$-estimation term we can use the same trick as in the beginning. Since the regularized loss is $\lambda$-strongly convex, we have that:

$$R_n(\widehat{\theta^{-j}}) \geq R_n(\theta^{\star}) + \nabla R_n(\theta^{\star})^{\top}(\widehat{\theta^{-j}} - \theta^{\star -j}).$$

Note that since $j \in \mathcal{H}_0$, we have that $z \perp\!\!\!\perp \chi^j \mid \chi^{-j}$ and therefore $\beta_j = 0$ (Prop. 5.2 from Reyero-Lobo et al. (2025b)). Thus, $\beta^{\star} = (0, \beta^{\star -j}) \in \{0\} \times \mathbb{R}^{p-1}$. Thus, since

$$\widehat{\theta^{-j}} = \operatorname{argmin}_{\theta \in \mathbb{R}^{p-1}} \frac{1}{n} \sum l(\theta^{\top} \chi_i^{-j}, z_i) + \lambda \|\theta\|^2,$$

we have in particular that $R_n(\widehat{\theta^{-j}}) \leq R_n(\theta^{\star})$. Then, using Cauchy-Schwarz as before, we simplify and obtain that

$$\|\widehat{\theta^{-j}} - \theta^{\star -j}\| \leq \frac{2}{\lambda} \|\nabla R_n(\theta^{\star})\|. \tag{10}$$

We observe that since

$$\theta^{\star} = \operatorname{argmin}_{\theta \in \mathbb{R}^p} R(\theta) := \operatorname{argmin}_{\theta \in \mathbb{R}^p} \mathbb{E}\left[l(\theta^{\top}\chi, z)\right] + \lambda \|\theta\|^2,$$

then $0 = \nabla R(\theta^{\star}) = \mathbb{E}\left[\chi \nabla l(\chi^{\top}\theta^{\star}, z)\right] + 2\lambda \theta^{\star}$. Thus, $2\lambda \theta^{\star} = -\mathbb{E}\left[\chi \nabla l(\chi^{\top}\theta^{\star}, z)\right]$. Also, note that

$$\nabla R_n(\theta^{\star}) = \frac{1}{n}\sum \chi_i \nabla l(\chi_i^{\top}\theta^{\star}, z_i) + 2\lambda \theta^{\star} = \frac{1}{n}\sum \chi_i \nabla l(\chi_i^{\top}\theta^{\star}, z_i) - \mathbb{E}\left[\chi \nabla l(\chi^{\top}\theta^{\star}, z)\right].$$

Therefore, combining this with (10), we have that

$$\|\Delta\| := \|\widehat{\theta^{-j}} - \theta^{\star -j}\| \leq \frac{2}{\lambda}\left\|\frac{1}{n}\sum \chi_i \nabla l(\chi_i^{\top}\theta^{\star}, z_i) - \mathbb{E}\left[\chi \nabla l(\chi^{\top}\theta^{\star}, z)\right]\right\|. \tag{11}$$

Thus, we have transformed the estimation bias into a centered sum to which we can apply the standard concentration inequalities.

Combining all the terms from (6), (7), (9), and finally (11)

$$
\begin{aligned}
\|\widehat{\theta} - \widetilde{\theta}\| &\leq \frac{2}{\lambda}\left|\frac{1}{n}\sum_{i=1}^{n}\chi_{ij}\left(\partial_u l({\theta^{\star -j}}^{\top}\chi_i^{-j}, z_i) + \left(\partial_u l(\widehat{\theta^{-j}}^{\top}\chi_i^{-j}, z_i) - \partial_u l({\theta^{\star -j}}^{\top}\chi_i^{-j}, z_i)\right)\right)\right| \\
&\leq \frac{2}{\lambda}\left|\frac{1}{n}\sum_{i=1}^{n}\chi_{ij}\partial_u l({\theta^{\star -j}}^{\top}\chi_i^{-j}, z_i)\right| + \frac{2}{\lambda}L\|M_j\|\|\Delta\| \\
&\leq \frac{2}{\lambda}\left|\frac{1}{n}\sum_{i=1}^{n}\chi_{ij}\partial_u l({\theta^{\star -j}}^{\top}\chi_i^{-j}, z_i)\right| + \frac{4L}{\lambda^2}\|M_j\|\left\|\frac{1}{n}\sum \chi_i \nabla l(\chi_i^{\top}\theta^{\star}, z_i) - \mathbb{E}\left[\chi \nabla l(\chi^{\top}\theta^{\star}, z)\right]\right\| \\
&\leq \frac{2}{\lambda}\left|\frac{1}{n}\sum_{i=1}^{n}\chi_{ij}s_i^{\star}\right| + \frac{4L}{\lambda^2}\|M_j\|\left\|\frac{1}{n}\sum \chi_i s_i^{\star} - \mathbb{E}\left[\chi s^{\star}\right]\right\|. 
\end{aligned}
\tag{12}
$$

Using Assumption E.3 (subgaussian score) and Assumption E.2 (subgaussian design), we have that the product is subexponential (see Lemma D.3): for $w_i := \chi_i^j s_i^{\star}$, $\|w_i\|_{\psi_1} \leq K_{\chi} K_s$.

Thus, using subexponential Bernstein (Theorem D.4),

$$\mathbb{P}\left(\left|\frac{1}{n}\sum_{i=1}^{n} w_i\right| \geq t\right) \leq 2\exp\left(-cn\min\left(\frac{t^2}{K_{\chi}^2 K_s^2}, \frac{t}{K_{\chi} K_s}\right)\right).$$

Working in the small-deviation regime ($t$ small enough),

$$\left| \frac{1}{n} \sum_{i=1}^{n} \chi_{ij} s_i^\star \right| \lesssim K_\chi K_s \sqrt{\frac{\log(2/\delta)}{n}}. \tag{13}$$

For the $M_j$ term, using Assumption E.2 we can bound this quantity in probability. First, as it is product of subgaussian, it is subexponential. Thus, we can apply Bernstein's inequality and in the small deviation regime, with small enough probability, we have

$$\|M_j - \Sigma_{-j,j}\|_2 \lesssim K_\chi^2 \sqrt{\frac{\log(1/\delta)}{n}}.$$

Since the population covariance operator norm is bounded by assumption, we have that with probability over $1 - \delta$,

$$\|M_j\|_2 \lesssim \|\Sigma_{-j,j}\|_2 + K_\chi^2 \sqrt{\frac{\log(1/\delta)}{n}} \leq B + O_P(1). \tag{14}$$

Finally, for the $\Delta$ term, we proceed similar as before but just taking into account the dimension. Indeed, denoting $G^l = \frac{1}{n}(\sum_i s_i^\star \chi_i^l - \mathbb{E}[s^\star \chi^l])$, by union bound we have that

$$\mathbb{P}\left(\max_l |G^l| \geq t\right) \leq 2(p-1)\exp\left(-cn\frac{t^2}{K_\chi^2 K_s^2}\right).$$

Thus, using that $\|G\|_2 \leq \sqrt{p-1}\max_l |G^l|$, we have that with probability $\geq 1 - \delta$,

$$\|G\|_2 \leq \sqrt{p-1}\frac{K_\chi K_s}{\sqrt{c}} \sqrt{\frac{\log(2(p-1)/\delta)}{n}}. \tag{15}$$

We finally conclude by directly plugging in (13), (14), and (15) into (12).

$\square$

### E.4. Semi-knockoffs $\mathcal{W}_1$-convergence: Theorem 4.2

**Theorem E.7** ($\mathcal{W}_1$ control)**.** *For $j \in \mathcal{H}_0$, under Theorem 4.1 assumptions and that model $\widehat{m}$ and loss in the prediction argument are Lipschitz, we have, with probability at least $1 - \delta$, that*

$$\mathcal{W}_1(\hat{P}_1^j, \hat{P}_2^j) \leq O_P\left(\sqrt{\frac{\log(1/\delta)}{n}}\right).$$

*Proof.* For the proof, we denote by $R$ the Lipschitz constant of the loss function and by $M$ the Lipschitz constant of the model $\widehat{m}$. We start by recalling the definition of the 1-Wasserstein distance between two distributions $\hat{P}_1$ and $\hat{P}_2$:

$$W_1(\hat{P}_1, \hat{P}_2) = \inf_{P \in \Theta(\hat{P}_1, \hat{P}_2)} \int_{\mathbb{R} \times \mathbb{R}} |x - y| P(\mathrm{dx}, \mathrm{dy}).$$

By the construction of the distributions from which we sample, and under the Lipschitz assumptions, we obtain

$$\begin{aligned}
W_1(\hat{P}_1, \hat{P}_2) &= \inf_{P \in \Theta(\hat{P}_1, \hat{P}_2)} \int_{\mathbb{R} \times \mathbb{R}} |x - y| P(\mathrm{dx}, \mathrm{dy}) \\
&= \inf_{P \in \Theta(\hat{P}_1, \hat{P}_2)} \int_{\mathbb{R}^p \times \mathbb{R} \times \mathbb{R}^p \times \mathbb{R}} |l(\widehat{m}(\widetilde{X}_1'), y_1) - l(\widehat{m}(\widetilde{X}_2'), y_2)| P(\mathrm{d}\widetilde{x}_1', \mathrm{dy}_1, \mathrm{d}\widetilde{x}_2', \mathrm{dy}_2) \\
&\leq RM \inf_{P \in \Theta(\hat{P}_1, \hat{P}_2)} \int_{\mathbb{R}^p \times \mathbb{R}^p} |\widetilde{X}_1' - \widetilde{X}_2'| P(\mathrm{d}\widetilde{x}_1', \mathrm{d}\widetilde{x}_2').
\end{aligned}$$

Since it is the infimum over all couplings, we use the information provided by the remaining coordinates from the same sample $(X_1, y_1)$, which are independent from those used for the residuals $(X_2, y_2)$ in both distributions; that is, we apply the same permutation to both empirical distributions:

$$
\begin{aligned}
W_1(\hat{P}_1, \hat{P}_2) &\leq RM \inf_{P \in \Theta(\hat{P}_1, \hat{P}_2)} \int_{\mathbb{R}^p \times \mathbb{R}^p} |\widetilde{X}'_1 - \widetilde{X}'_2| P(\mathrm{d}\widetilde{\mathrm{x}}'_1, \mathrm{d}\widetilde{\mathrm{x}}'_2) \\
&\leq RM \int_{\mathbb{R}^p \times \mathbb{R}^p} \left| \left( \widehat{\nu}(X_1^{-j}) + \left( X_2^j - \widehat{\nu}(X_2^{-j}) \right) \right) \right. \\
&\qquad \left. - \left( \widehat{\rho}(X_1^{-j}, y_1) + \left( X_2^j - \widehat{\rho}(X_2^{-j}, y_2) \right) \right) \right| P(\mathrm{dx}_1^{-j}, \mathrm{dy}_1) \times P(\mathrm{dx}_2, \mathrm{dy}_2) \\
&= RM \int_{\mathbb{R}^p \times \mathbb{R}^p} \left| \left( \widehat{\nu}(X_1^{-j}) - \widehat{\rho}(X_1^{-j}, y_1) \right) \right. \\
&\qquad \left. - \left( \widehat{\nu}(X_2^{-j}) - \widehat{\rho}(X_2^{-j}, y_2) \right) \right| P(\mathrm{dx}_1^{-j}, \mathrm{dy}_1) \times P(\mathrm{dx}_2^{-j}, \mathrm{dy}_2) \\
&\leq 2RM \int_{\mathbb{R}^p} \left| \left( \widehat{\nu}(X^{-j}) - \widehat{\rho}(X^{-j}, y) \right) \right| P(\mathrm{dx}^{-j}, \mathrm{dy}).
\end{aligned}
$$

Note that both $\widehat{\nu}$ and $\widehat{\rho}$ are linear, since they are defined as

$$
\widehat{\nu}(X^{-j}) := \widehat{\theta^{-j}}^{\top} X^{-j} \quad \text{and} \quad \widehat{\rho}(X^{-j}, y) := \widehat{\theta}^{\top}(X^{-j}, y).
$$

Similarly to the stability section, we denote by $\widetilde{\theta}$ the vector $\widehat{\theta^{-j}}$ extended by a $0$ for the $y$-entry, so that both vectors have the same dimension. Thus, we have that

$$
\begin{aligned}
W_1(\hat{P}_1, \hat{P}_2) &\leq 2RM \int_{\mathbb{R}^p} \left| \left( \widehat{\nu}(X^{-j}) - \widehat{\rho}(X^{-j}, y) \right) \right| P(\mathrm{dx}^{-j}, \mathrm{dy}) \\
&= 2RM \int_{\mathbb{R}^p} \left| \left( \widehat{\theta^{-j}}^{\top} X^{-j} - \widehat{\theta}^{\top}(X^{-j}, y) \right) \right| P(\mathrm{dx}^{-j}, \mathrm{dy}) \\
&= 2RM \int_{\mathbb{R}^p} \left| \left( \widetilde{\theta}^{\top}(X^{-j}, y) - \widehat{\theta}^{\top}(X^{-j}, y) \right) \right| P(\mathrm{dx}^{-j}, \mathrm{dy}) \\
&\leq 2RM \int_{\mathbb{R}^p} \left\| \widetilde{\theta} - \widehat{\theta} \right\| \left\| (X^{-j}, y) \right\| P(\mathrm{dx}^{-j}, \mathrm{dy}) \\
&\lesssim O_p \left( \sqrt{\frac{\log(1/\delta)}{n}} \right) \int_{\mathbb{R}^p} \left\| (X^{-j}, y) \right\| P(\mathrm{dx}^{-j}, \mathrm{dy}),
\end{aligned}
$$

where for the last step we used Theorem 4.1. To conclude, we just need to observe that the last expectation is bounded, which is the case from the subgaussian assumption. $\square$

### E.5. Double Robustness: Theorem 4.3

**Theorem E.8** (Double Robustness). *Assume that $j \in \mathcal{H}_0$ and that $t \to l(t, y)$ is $C^2$ with bounded derivatives. Assume also that $\widehat{m}$ is differentiable in the $j$-coordinate and denote by $a_n := \partial_{x^j} \widehat{m}(X^{-j}, t)$ for a $t$ between $\widetilde{X}'^j$ and $\widetilde{X}^j$. Assume that $\widehat{\nu} - \nu = O_P(b_n)$. Then,*

$$
l(\widehat{m}(\widetilde{X}'), y) - l(\widehat{m}(\widetilde{X}), y) = O_P(a_n b_n).
$$

*Proof.* We begin by applying Taylor to $t \to l(t, y)$. We obtain

$$
l(\widehat{m}(\widetilde{X}'), y) - l(\widehat{m}(\widetilde{X}), y) = l'(\widehat{m}(\widetilde{X}), y)(\widehat{m}(\widetilde{X}') - \widehat{m}(\widetilde{X})) + \frac{1}{2} l''(u, y)(\widehat{m}(\widetilde{X}') - \widehat{m}(\widetilde{X}))^2, \tag{16}
$$

for some $u$ between $\widehat{m}(\widetilde{X}')$ and $\widehat{m}(\widetilde{X})$. Now, applying the mean-value theorem to $t \to \widehat{m}(X^{-j}, t)$ we have that

$$
\begin{aligned}
\widehat{m}(\widetilde{X}') - \widehat{m}(\widetilde{X}) &= \partial_{x^j} \widehat{m}(X^{-j}, t)(\widetilde{X}'^j - \widetilde{X}^j) \\
&= a_n \left( \left( \widehat{\nu}(X^{-j}) + (X'^j - \widehat{\nu}(X'^{-j})) \right) - \left( \nu(X^{-j}) + (X'^j - \nu(X'^{-j})) \right) \right) \\
&= a_n \left( \left( \widehat{\nu}(X^{-j}) - \nu(X^{-j}) \right) + \left( \widehat{\nu}(X'^{-j}) - \nu(X'^{-j}) \right) \right) \\
&= O_P(a_n b_n).
\end{aligned}
$$

Finally, we obtain the result by plugging the last into (16) and using the boundedness of the loss.

$\square$

From the proof, we also observe that the assumptions on the loss function could be relaxed to merely require Lipschitz continuity instead of doing the Taylor expansion in (16).

## F. Additional experiments

### F.1. Summary of methods

In this section, we provide a brief summary of standard procedures for Conditional Independence Testing, drawn from both Interpretable Machine Learning (IML) / Variable Importance Measures (VIM) and Controlled Variable Selection. In the main text, for brevity, we abbreviate names (e.g., Wcx for Wilcoxon, SKO for Semi-knockoffs, SCPI for Sobol-CPI, and p for the number of permutations used in derandomization). Here, we revert to the original terminology.

Although our experiments include all methods listed in Table 2, for readability we display only the most relevant methods in the figures.

| Method | Reference | Guarantee |
|---|---|---|
| Sobol-CPI(1/10/100) | (Chamma et al., 2024) (Watson & Wright, 2021) (Covert et al., 2020) | None |
| Sobol-CPI(1/10/100)_sqrt | (Reyero-Lobo et al., 2025a) | Asympt. type-I error |
| Sobol-CPI(1/10/100)_n / bt | (Reyero-Lobo et al., 2025a) | Asympt. type-I error under linear settings |
| Sobol-CPI(1/10/100)_n2 | - | None |
| Sobol-CPI(1)_ST | (Watson & Wright, 2021) (Reyero-Lobo et al., 2025a) | Type-I error |
| Sobol-CPI(1)_Wilcox | (Reyero-Lobo et al., 2025a) | Type-I error |
| LOCO-W | (Williamson et al., 2021) | Asympt. type-I error |
| LOCO | - | None |
| LOCO_sqrt | (Verdinelli & Wasserman, 2024) | Asympt. type-I error |
| LOCO_n/_n2 | - | None |
| LOCO_ST/_Wilcox | (Lei et al., 2018) | Type-I error |
| dCRT | (Liu et al., 2022) | Type-I error |
| HRT | (Tansey et al., 2022) | Type-I error |
| Knockoffs | (Candès et al., 2018) | FDR |
| Semi_KO | Ours | FDR |
| Semi_KO_ST/_Wilcox | Ours | Type-I error |

*Table 2.* Summary of Conditional Independence Testing procedures using Machine Learning methods.

### F.2. Double Robustness

In this section, we provide additional insight into the double robustness phenomenon illustrated in Figure 3. Our goal is to show that the decay obtained when both the imputer and the model coordinate to detect a null feature is faster than

$$\mathcal{D}_{\text{train},\,\hat{m}} \neq \mathcal{D}_{\text{train},\,\hat{\nu}} \neq \mathcal{D}_{\text{test}}$$

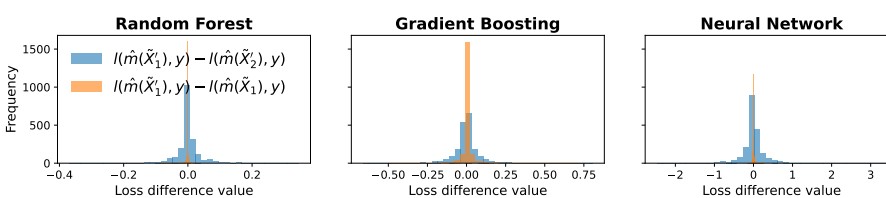

*Figure 7.* **Double Robustness:** Distribution of the semi-knockoff statistic, i.e., at two independently sampled estimated residuals (blue: $l(\widehat{m}(\widetilde{X}_1'),y) - l(\widehat{m}(\widetilde{X}_2'),y)$), and distribution of the difference between the theoretical and estimated residuals (orange: $l(\widehat{m}(\widetilde{X}_1'),y) - l(\widehat{m}(\widetilde{X}_1),y)$) for a null coordinate ($j = 0$). The data-generating model is $y = 0.8X^1 + 0.6X^2 + 0.4X^3 + 0.2X^4 + \sin(X^1) + \epsilon$, with $\epsilon \sim \mathcal{N}(0, 0.5)$. We use $n = 2000$ samples with $X \sim \mathcal{N}(0, \Sigma)$, where $\Sigma^{i,j} = 0.5^{|i-j|}$. We use an independent test set and independent train set for each model.

$$\mathcal{D}_{\text{train},\,\hat{m}} = \mathcal{D}_{\text{train},\,\hat{\nu}} \neq \mathcal{D}_{\text{test}}$$

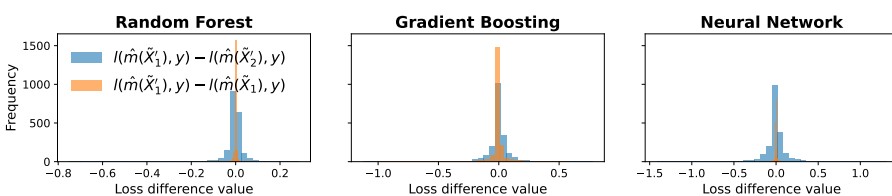

*Figure 8.* **Double Robustness:** Distribution of the semi-knockoff statistic, i.e., the difference in loss evaluated at two independently sampled estimated residuals (blue: $l(\widehat{m}(\widetilde{X}_1'),y) - l(\widehat{m}(\widetilde{X}_2'),y)$), and distribution of the difference between the theoretical and estimated residuals (orange: $l(\widehat{m}(\widetilde{X}_1'),y) - l(\widehat{m}(\widetilde{X}_1),y)$) for a null coordinate ($j = 0$). The data-generating model is $y = 0.8X^1 + 0.6X^2 + 0.4X^3 + 0.2X^4 + \sin(X^1) + \epsilon$, with $\epsilon \sim \mathcal{N}(0, 0.5)$. We use $n = 2000$ samples with $X \sim \mathcal{N}(0, \Sigma)$, where $\Sigma^{i,j} = 0.5^{|i-j|}$. We use an independent test set and the same train set for each model.

the decay achieved by the model alone. This is visualized by the blue histogram, which is constructed by evaluating the difference in model outputs across two independent imputations (i.e., the residuals are perturbed by two independent random permutations). This distribution captures the variability that comes exclusively from the model, which does not rely on the null feature since it has no predictive power.

In contrast, the orange distribution compares, for the same model, the performance of an estimated imputer with that of the oracle imputer under the same permutation (and thus the same residual perturbations). Since both the model and the imputer cooperate to detect the null feature, the convergence toward zero is faster, resulting in the orange distribution being more concentrated around zero than the blue distribution obtained from independent estimated residuals. This observation underlies the conjecture described in Section 4.5 and in Figure 2.

Moreover, it is important to note that across all experiments and models, the blue distribution—which is the one used in practice—is centered and approximately symmetric around zero, thereby guaranteeing exchangeability and, in turn, practical FDR control.

In Figures 7, 8, and 9, we present the experiments in three different data-splitting regimes: (i) the test sample on which the model is evaluated is independent from both the training sample used to fit $\widehat{m}$ and the sample used to train the imputer $\widehat{\nu}$, which are also independent from each other; (ii) only the training and test samples are independent; and (iii) the same dataset is used for training $\widehat{m}$, training $\widehat{\nu}$, and evaluation. We observe that the behavior is qualitatively similar across these regimes and that the double robustness phenomenon persists. These experiments were conducted using Random Forest, Gradient Boosting, and Neural Network models.

Finally, in Figures 10 and 11, we study the same property in a higher-dimensional setting and include additional models. We observe that the double robustness phenomenon persists, as indicated by the orange distribution being more concentrated around zero, and that exchangeability still holds in practice, since the blue distribution remains approximately symmetric around zero.

$$\mathcal{D}_{\text{train}, \hat{m}} = \mathcal{D}_{\text{train}, \hat{\nu}} = \mathcal{D}_{\text{test}}$$

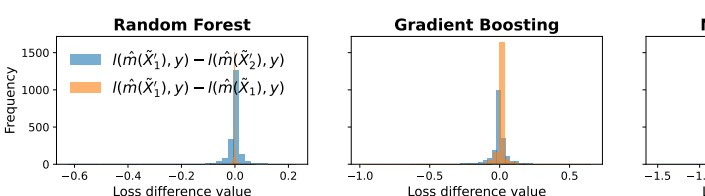

*Figure 9.* **Double Robustness:** Distribution of the semi-knockoff statistic, i.e., the difference in loss evaluated at two independently sampled estimated residuals (blue: $l(\widehat{m}(\widetilde{X}_1'), y) - l(\widehat{m}(\widetilde{X}_2'), y)$), and distribution of the difference between the theoretical and estimated residuals (orange: $l(\widehat{m}(\widetilde{X}_1'), y) - l(\widehat{m}(\widetilde{X}_1), y)$) for a null coordinate ($j = 0$). The data-generating model is $y = 0.8X^1 + 0.6X^2 + 0.4X^3 + 0.2X^4 + \sin(X^1) + \epsilon$, with $\epsilon \sim \mathcal{N}(0, 0.5)$. We use $n = 2000$ samples with $X \sim \mathcal{N}(0, \Sigma)$, where $\Sigma^{i,j} = 0.5^{|i-j|}$. We use a unique test and train set.

$$\mathcal{D}_{\text{train}, \hat{m}} \neq \mathcal{D}_{\text{train}, \hat{\nu}} \neq \mathcal{D}_{\text{test}}$$

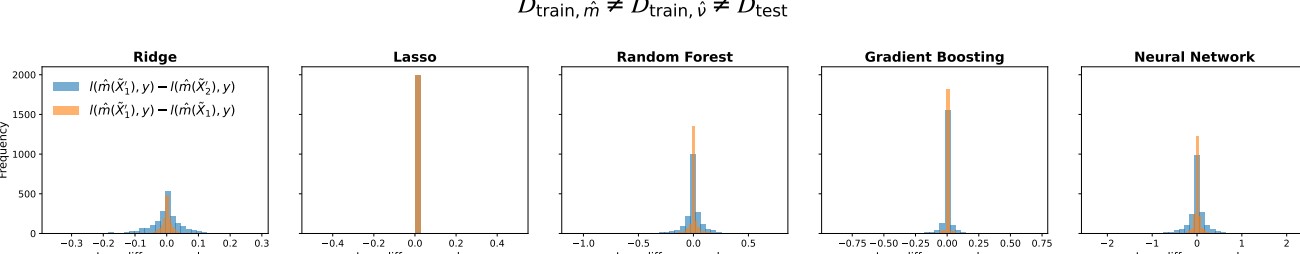

*Figure 10.* **Double Robustness:** Distribution of the semi-knockoff statistic, i.e., the difference in loss evaluated at two independently sampled estimated residuals (blue: $l(\widehat{m}(\widetilde{X}_1'), y) - l(\widehat{m}(\widetilde{X}_2'), y)$), and distribution of the difference between the theoretical and estimated residuals (orange: $l(\widehat{m}(\widetilde{X}_1'), y) - l(\widehat{m}(\widetilde{X}_1), y)$) for a null coordinate ($j = 0$). The data-generating model is $y = X^\top \beta + \epsilon$, with $\epsilon \sim \mathcal{N}(0, 0.5)$ and $\beta$ 0.5-sparse randomly sampled with half of the values equally sampled from 0.3 to 1, and the other half from $-0.8$ to $-0.2$. We use $n_{\text{train},,\widehat{m}} = 500$ samples for training $\widehat{m}$ and $n_{\text{train},\widehat{\nu}} = n_{\text{test}} = 2000$ for training $\widehat{\nu}$ and $\widehat{\rho}$ with $X \sim \mathcal{N}(0, \Sigma)$, where $\Sigma^{i,j} = 0.5^{|i-j|}$ and $X \in \mathbb{R}^{50}$. We use an independent test set and independent train set for each model.

$$\mathcal{D}_{\text{train}, \hat{m}} = \mathcal{D}_{\text{train}, \hat{\nu}} = \mathcal{D}_{\text{test}}$$

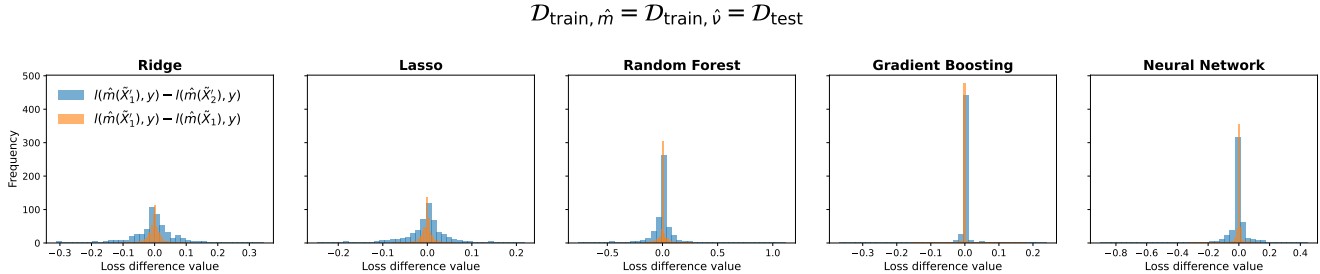

*Figure 11.* **Double Robustness:** Distribution of the semi-knockoff statistic, i.e., the difference in loss evaluated at two independently sampled estimated residuals (blue: $l(\widehat{m}(\widetilde{X}_1'), y) - l(\widehat{m}(\widetilde{X}_2'), y)$), and distribution of the difference between the theoretical and estimated residuals (orange: $l(\widehat{m}(\widetilde{X}_1'), y) - l(\widehat{m}(\widetilde{X}_1), y)$) for a null coordinate ($j = 0$). The data-generating model is $y = X^\top \beta + \epsilon$, with $\epsilon \sim \mathcal{N}(0, 0.5)$ and $\beta$ 0.5-sparse randomly sampled with half of the values equally sampled from 0.3 to 1, and the other half from $-0.8$ to $-0.2$. We use $n = 500$ samples with $X \sim \mathcal{N}(0, \Sigma)$, where $\Sigma^{i,j} = 0.5^{|i-j|}$ and $X \in \mathbb{R}^{50}$. We use a unique test and train set.

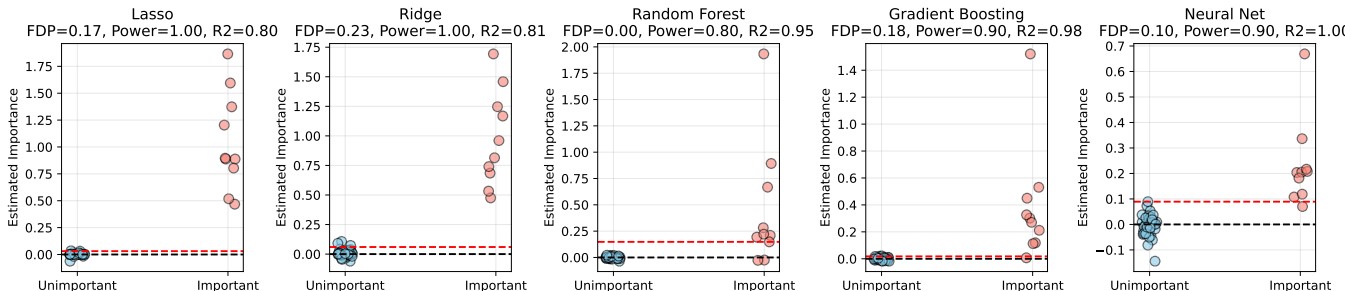

*Figure 12.* **Semi-knockoff statistics:** $y = X\beta + \epsilon$, where $\beta$ is 0.25-sparse with important features grouped in blocks of 5 sampled uniformly. The noise level is $\|X\beta\|/2$. The design matrix satisfies $X \sim \mathcal{N}(0, \Sigma)$ with $\Sigma_{i,j} = 0.6^{|i-j|}$. We set $n = 300$ and $p = 50$. The figure displays the distribution of the Semi-knockoff statistics for several standard ML models. In each plot, the null features are shown on the left and the important features on the right. The red horizontal line denotes the data-dependent knockoff threshold corresponding to a target FDR level of $q = 0.2$.

## F.3. Exchangeability

In Figure 12, we observe that, in practice, the sampling scheme effectively preserves the required symmetry under the null. This exchangeability (Lemma 2.1) lies at the heart of knockoff FDR control, since the threshold estimates the number of false discoveries by exploiting this symmetry. This is reflected by the distribution of the null features (blue dots), which is approximately symmetric around 0. As a result, the knockoff threshold successfully controls the FDR across standard ML models, such as Lasso, Ridge, Gradient Boosting, Random Forest, and Neural Networks. Moreover, the important features exhibit a distribution that is clearly separated from that of the nulls, yielding a powerful testing procedure that consists of selecting the coordinates whose statistics are above the threshold (red line).

## F.4. Type-I error

In this section, we present the type-I error control experiments using the settings from the main text across several models: the adjacent support setting from Figure 4 and the masked correlation setting from Figure 5. Our goal is to verify that the observed behavior remains consistent for standard ML models, namely neural networks, gradient boosting, and random forests. All models were implemented using `scikit-learn` (Pedregosa et al., 2011). We recall that across the experiments $n = 300$ and $p = 50$. We additionally include an analysis of computation time.

**Computation time.**  Semi-knockoffs are expected to offer substantial computational improvements relative to other CIT methods such as LOCO, dCRT, and HRT. In our experiments, we used multiple models with default hyperparameters, so the differences were somewhat muted for complex models. However, LOCO and the distillation step of dCRT require re-fitting a complex model for each feature. Moreover, both dCRT and HRT require resampling the data thousands of times to compute each p-value. In contrast, Semi-knockoffs can be optimized to avoid these repeated refits and resampling steps, and therefore have the potential to be significantly faster in practice.

### F.4.1. ADJACENT SUPPORT

This setting consists of having an adjacent support, such that all important features $(0.25p)$ form a contiguous block. The results were similar with a randomly sampled support. The main observations from Figure 4 are: (i) the low power of VIM, which highlights the need for a CIT procedure compatible with arbitrary pre-trained models; and (ii) the reduced power of HRT due to the data-splitting step, relative to Semi-knockoffs. These conclusions are consistent with Figures 13, 14, and 15, corresponding to neural networks, gradient boosting, and random forests, respectively. The only exception is the neural network case, where the model attains high predictive accuracy even under data splitting ($R^2 = 0.963$), resulting in HRT exhibiting full power and therefore not lagging behind methods that avoid data splitting ($R^2 = 0.965$). For the other models, we observe a clear gain from avoiding the data-splitting step.

Overall, all methods control the type-I error except `LOCO-W` (Williamson et al., 2023), which provides only asymptotic type-I error guarantees and requires an additional train–train split for the original and restricted models; in practice, it fails to control the type-I error and also incurs a loss in power. We also note that, similarly to the power, the AUC for `dCRT` and `Semi-knockoffs` outperforms the other methods, and that a 5-permutation derandomization step can be beneficial for

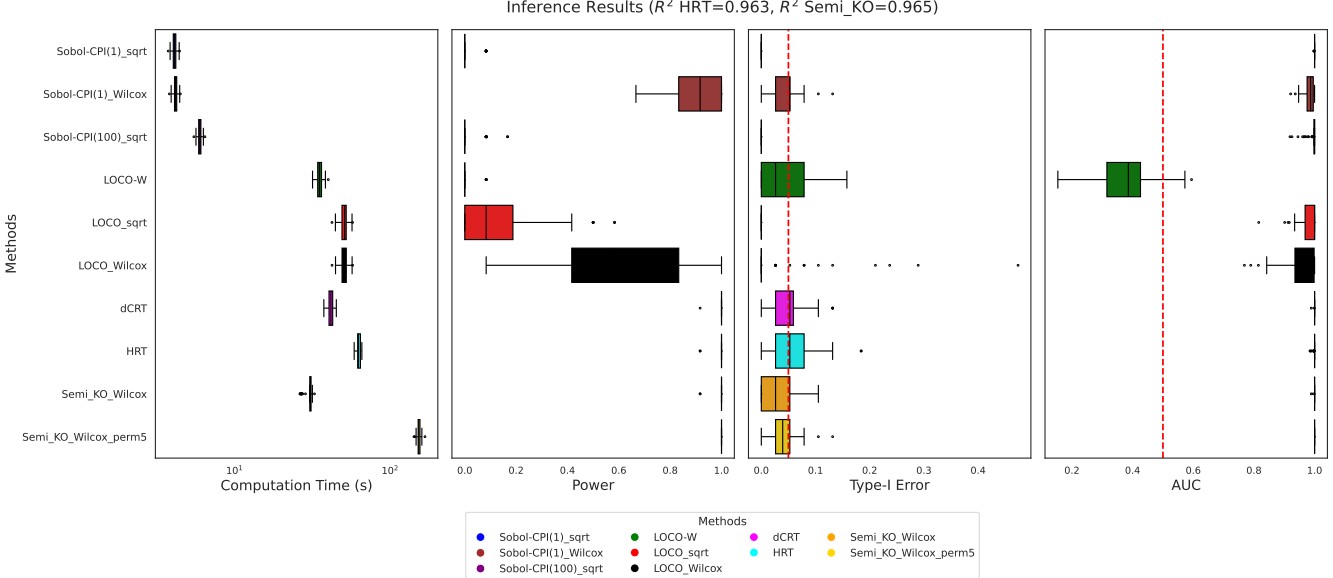

*Figure 13.* **Type-I error adjacent support with Neural Network.** Let $X \sim \mathcal{N}(0, \Sigma)$ with $\Sigma^{ij} = 0.6^{|i-j|}$, and $y = \beta^\top X + \epsilon$, where the first $0.25p$ coordinates of $\beta$ are between 1 and 2 and the remaining entries are zero, and $\epsilon \sim \mathcal{N}(0, 1)$. The black-box pretrained model $\widehat{m}$ is a neural network.

power.

### F.4.2. MASKED CORRELATION

This setting consists of a single important coordinate that is highly correlated with an unimportant feature. Under this configuration, we observe a clear gain from using a pretrained model that can evaluate the contribution of all features jointly. In contrast, the distillation step in dCRT—which consists of training a complex model between $y$ and $X^{-j}$ in order to isolate the importance of feature $X^j$ from the residual—leads to a loss of power, since the distillation shrinks the signal provided by the important feature.

The conclusions here are similar to those in the previous setting. Specifically, `Sobol-CPI` and `LOCO` (except for the Wilcoxon version) exhibit no power due to overly restrictive corrections, and there is also a noticeable loss from data splitting in `HRT`. Finally, the derandomization step increases the power of `Semi-knockoffs`.

### F.5. False Discovery Rate

In this section, we provide additional intuition regarding the FDR control experiments for Semi-knockoffs. All previously discussed procedures that produce valid p-values can be adapted to FDR control by applying a multiple-testing correction procedure such as BH (Benjamini & Hochberg, 1995). However, such guarantees are not exact, since the dependence structure among the p-values is unknown, and may violate the assumptions under which BH ensures FDR control.

In contrast, procedures such as standard `Knockoffs` and `Semi-knockoffs` provide exact FDR control by construction. In principle, the Semi-knockoffs framework could be adapted either by applying BH to the p-values from the Wilcoxon version, or by directly applying the knockoff threshold to control the FDR. As our experiments indicate, the latter strategy should be preferred in practice and in theory.

We present results for the Adjacent setting from the previous section, but now focusing on FDR control, as well as for a heavy-tailed setting. These experiments are conducted using neural network, gradient boosting, and random forest models.

### F.5.1. ADJACENT SUPPORT

We revisit the setting with a block of important features. Since the underlying model is linear, `Knockoffs` achieves high power, as expected. Moreover, we observe a similar pattern to that seen in the p-value experiments: for gradient boosting

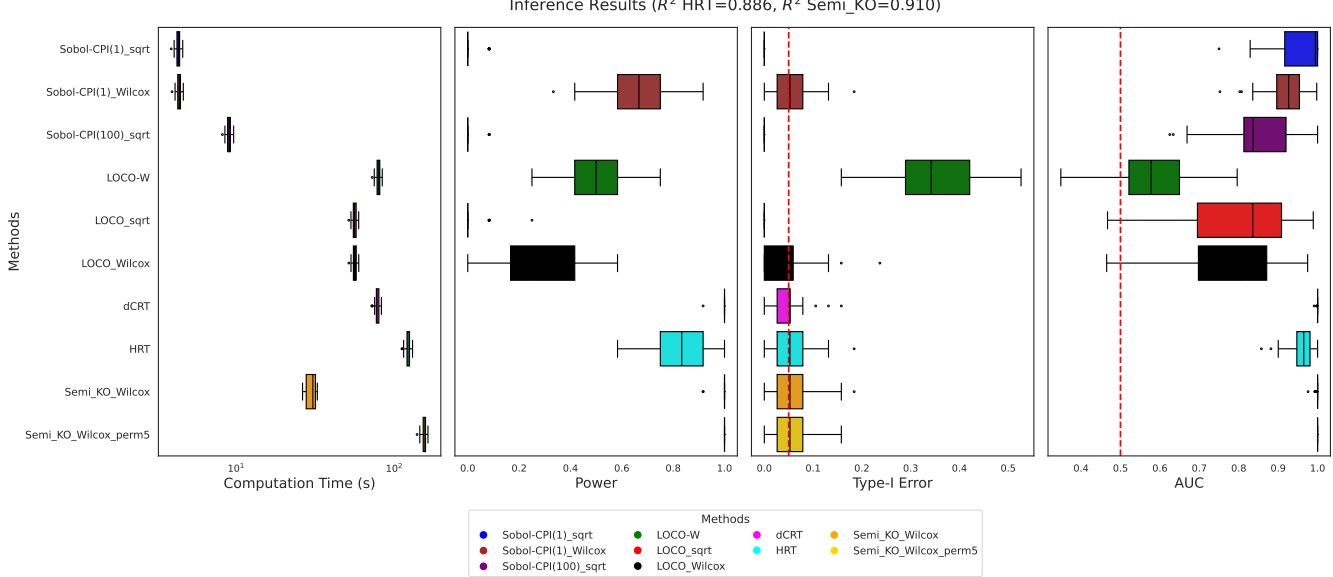

*Figure 14.* **Type-I error adjacent support with Gradient Boosting.** Let $X \sim \mathcal{N}(0, \Sigma)$ with $\Sigma^{ij} = 0.6^{|i-j|}$, and $y = \beta^\top X + \epsilon$, where the first $0.25p$ coordinates of $\beta$ are between 1 and 2 and the remaining entries are zero, and $\epsilon \sim \mathcal{N}(0, 1)$. The black-box pretrained model $\widehat{m}$ is a gradient boosting.

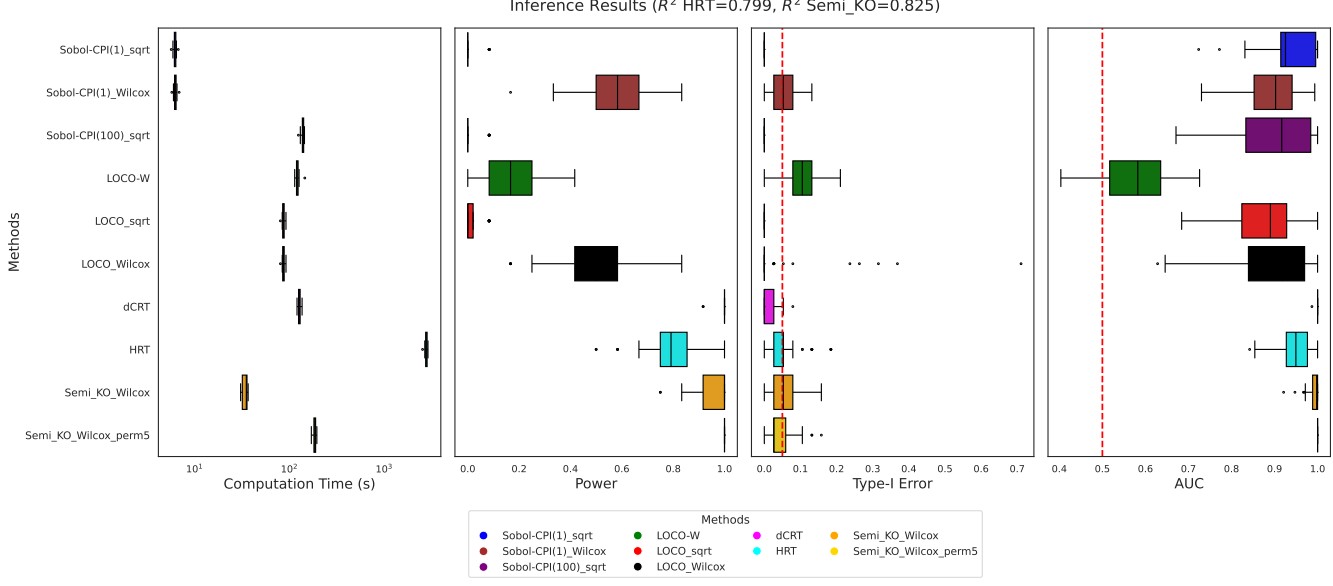

*Figure 15.* **Type-I error adjacent support with Random Forest.** Let $X \sim \mathcal{N}(0, \Sigma)$ with $\Sigma^{ij} = 0.6^{|i-j|}$, and $y = \beta^\top X + \epsilon$, where the first $0.25p$ coordinates of $\beta$ are between 1 and 2 and the remaining entries are zero, and $\epsilon \sim \mathcal{N}(0, 1)$. The black-box pretrained model $\widehat{m}$ is a random forest.

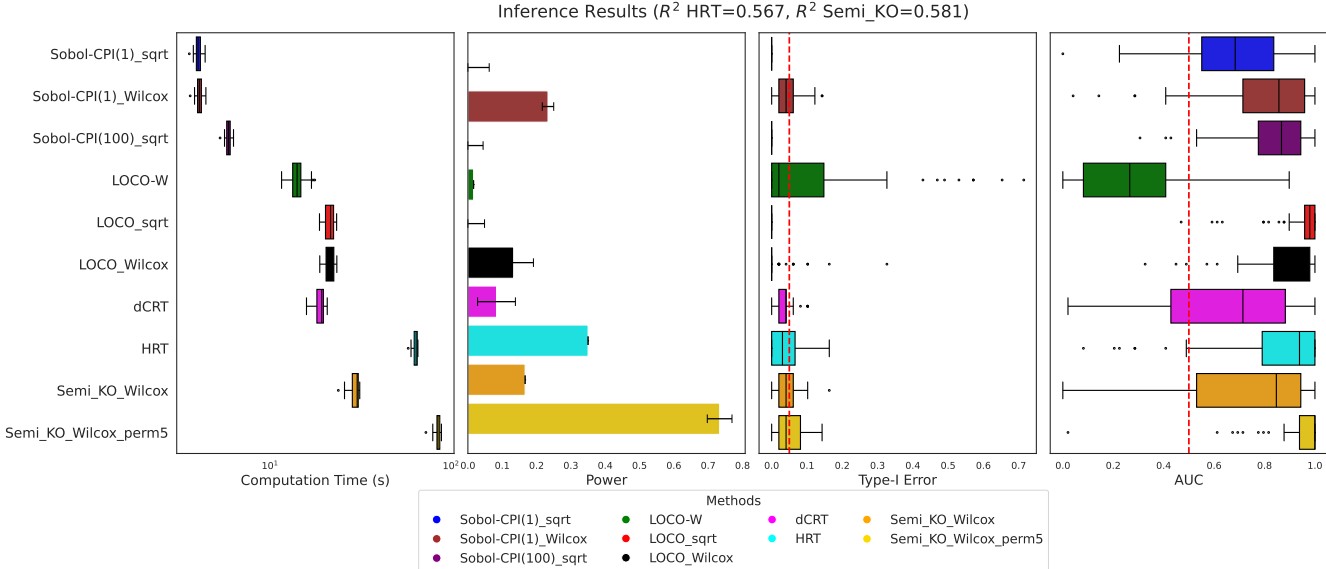

*Figure 16.* **Type-I error masked correlation with Neural Network.** Let $X \sim \mathcal{N}(0, \Sigma)$ with $\Sigma_{ij} = 0.6^{|i-j|}$. A unique relevant coordinate $l$ is sampled and $y = X_l + 0.5\,\epsilon_1$, where $\epsilon_1 \sim \mathcal{N}(0, 1)$. A correlated null variable is created as $X_{l-1} = X_l + 0.5\,\epsilon_2$, with $\epsilon_2 \sim \mathcal{N}(0, 1)$. The black-box pretrained model $\widehat{m}$ is a neural network.

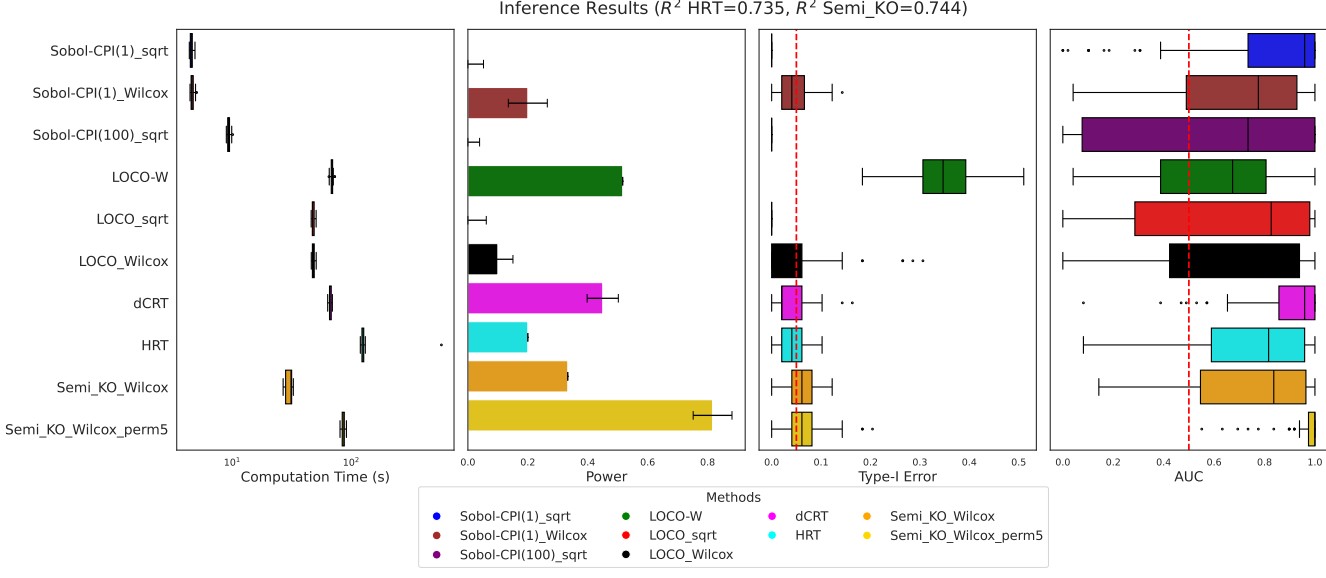

*Figure 17.* **Type-I error masked correlation with Gradient Boosting.** Let $X \sim \mathcal{N}(0, \Sigma)$ with $\Sigma_{ij} = 0.6^{|i-j|}$. A unique relevant coordinate $l$ is sampled and $y = X_l + 0.5\,\epsilon_1$, where $\epsilon_1 \sim \mathcal{N}(0, 1)$. A correlated null variable is created as $X_{l-1} = X_l + 0.5\,\epsilon_2$, with $\epsilon_2 \sim \mathcal{N}(0, 1)$. The black-box pretrained model $\widehat{m}$ is a gradient boosting.

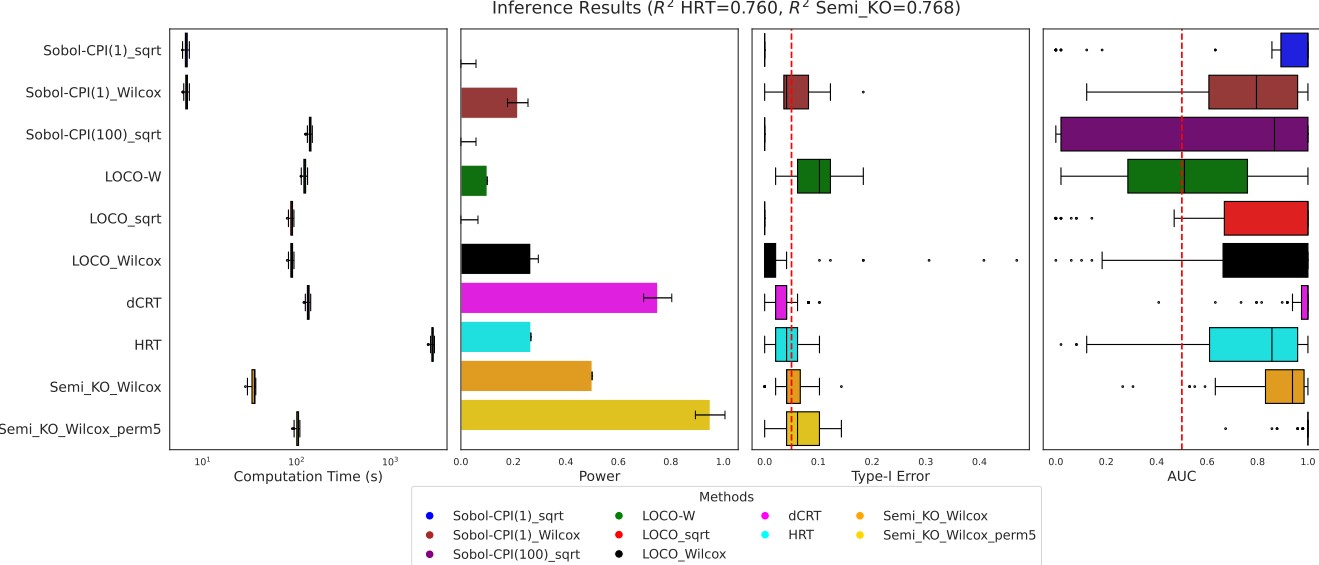

*Figure 18.* **Type-I error masked correlation with Random Forest.** Let $X \sim \mathcal{N}(0, \Sigma)$ with $\Sigma_{ij} = 0.6^{|i-j|}$. A unique relevant coordinate $l$ is sampled and $y = X_l + 0.5\,\epsilon_1$, where $\epsilon_1 \sim \mathcal{N}(0,1)$. A correlated null variable is created as $X_{l-1} = X_l + 0.5\,\epsilon_2$, with $\epsilon_2 \sim \mathcal{N}(0,1)$. The black-box pretrained model $\widehat{m}$ is a random forest.

(Figure 20) and random forest (Figure 21), there is a clear gain from avoiding the train–test split. In particular, HRT does not attain full power, which can be attributed to the reduction in $R^2$ caused by data splitting.

### F.5.2. HEAVY-TAILS

In this setting, the input features are no longer Gaussian. As a result, there is a reduced predictive performance compared to the adjacent setting. For instance, the $R^2$ of the neural network decreases from $R^2 = 0.965$ (with splitting: $R^2 = 0.964$) to $R^2 = 0.828$ (with splitting: $R^2 = 0.798$), and for gradient boosting from $R^2 = 0.910$ (with splitting: $R^2 = 0.884$) to $R^2 = 0.737$ (with splitting: $R^2 = 0.688$).

Nevertheless, we still observe that HRT loses power due to the train–test split. Overall, it is preferable to use `Semi-knockoffs` for FDR control rather than applying BH to the `Semi-Knockoffs-Wilcoxon` p-values, and the derandomization procedure further increases power. Note that in this linear setting, the knockoffs are powerful since they rely on the Lasso coefficient difference. However, Gaussian knockoff variables are no longer valid here, so the FDR is not theoretically controlled.

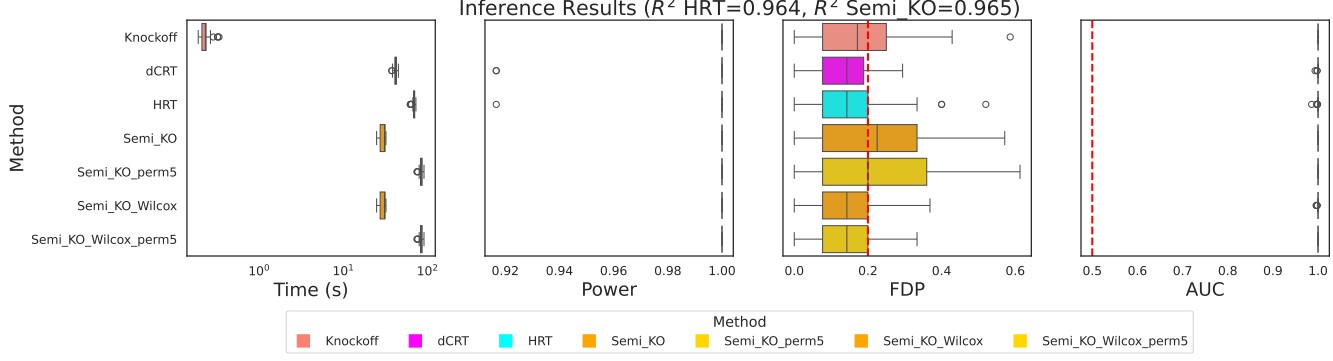

*Figure 19.* **FDR adjacent support with Neural Network.** Let $X \sim \mathcal{N}(0, \Sigma)$ with $\Sigma^{ij} = 0.6^{|i-j|}$, and $y = \beta^\top X + \epsilon$, where the first $0.25p$ coordinates of $\beta$ are between 1 and 2 and the remaining entries are zero, and $\epsilon \sim \mathcal{N}(0,1)$. The black-box pretrained model $\widehat{m}$ is a neural network, achieving $R^2 = 0.964$ with a train-test split and $R^2 = 0.965$ without split.

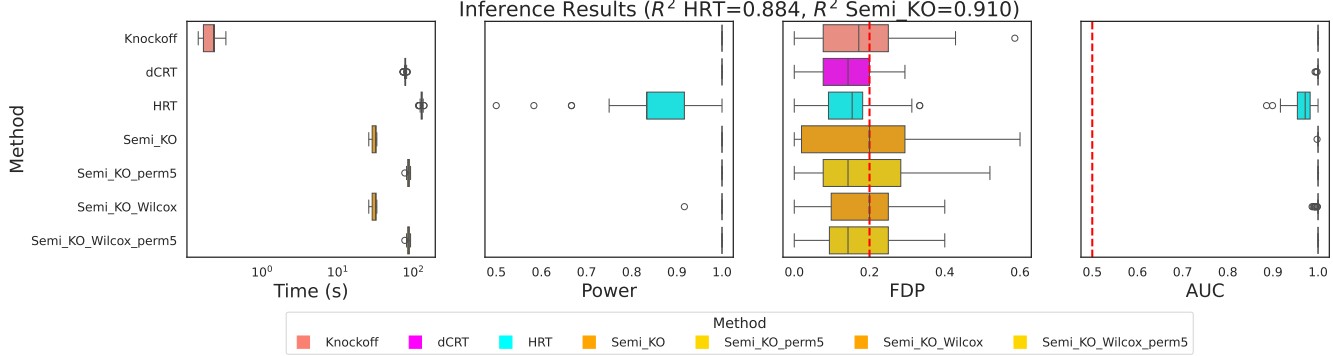

*Figure 20.* **FDR adjacent support with Gradient Boosting.** Let $X \sim \mathcal{N}(0, \Sigma)$ with $\Sigma^{ij} = 0.6^{|i-j|}$, and $y = \beta^{\top} X + \epsilon$, where the first $0.25p$ coordinates of $\beta$ are between 1 and 2 and the remaining entries are zero, and $\epsilon \sim \mathcal{N}(0, 1)$. The black-box pretrained model $\widehat{m}$ is a gradient boosting, achieving $R^2 = 0.884$ with a train-test split and $R^2 = 0.910$ without split.

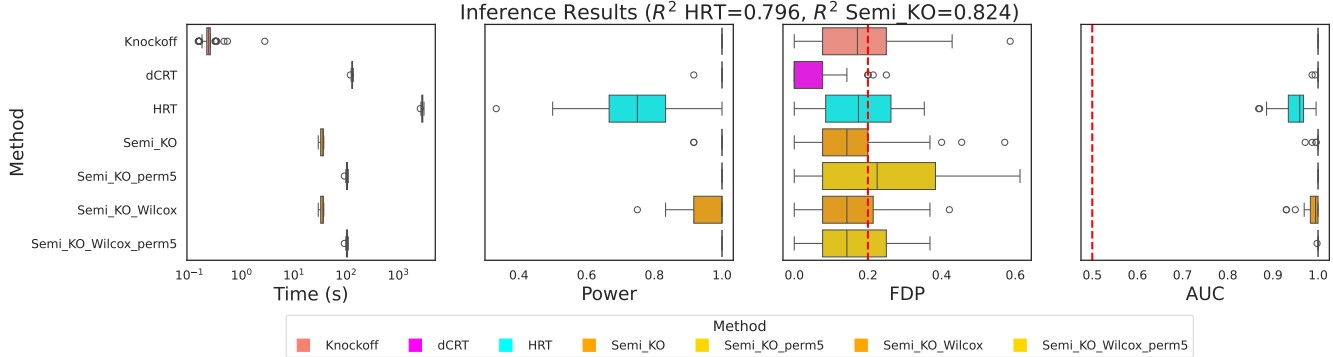

*Figure 21.* **FDR adjacent support with Random Forest.** Let $X \sim \mathcal{N}(0, \Sigma)$ with $\Sigma^{ij} = 0.6^{|i-j|}$, and $y = \beta^{\top} X + \epsilon$, where the first $0.25p$ coordinates of $\beta$ are between 1 and 2 and the remaining entries are zero, and $\epsilon \sim \mathcal{N}(0, 1)$. The black-box pretrained model $\widehat{m}$ is a random forest, achieving $R^2 = 0.796$ with a train-test split and $R^2 = 0.824$ without split.

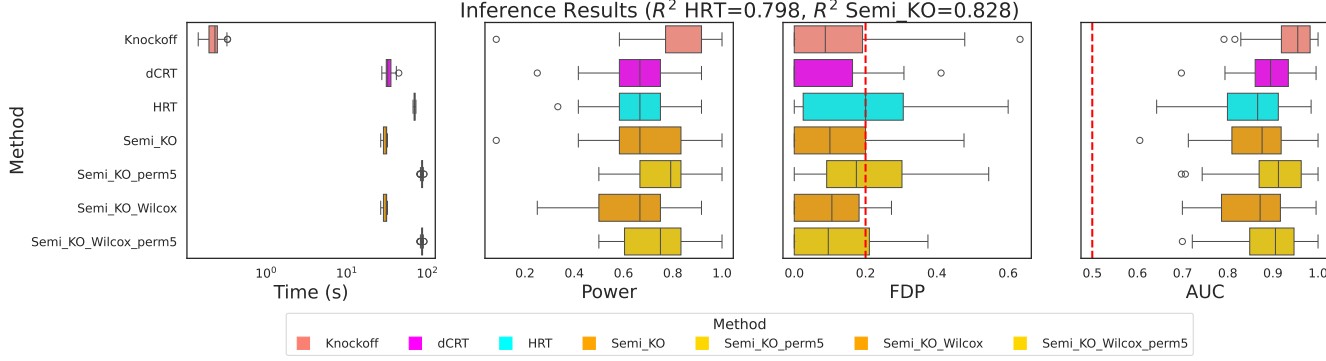

*Figure 22.* **FDR heavy-tailed with Neural Networks.** Let $X = \Sigma^{1/2} Z$, where $Z$ has i.i.d. t-Student entries $Z_{ij} \sim t_3$ and $\Sigma_{ij} = 0.6^{|i-j|}$. We generate $y = \beta^{\top} X + \epsilon$, where the first $0.25p$ coordinates of $\beta$ lie in $[1, 2]$ and the remaining entries are zero, with $\epsilon \sim \mathcal{N}(0, 1)$. The black-box pretrained model $\widehat{m}$ is a neural network, achieving $R^2 = 0.798$ with a train-test split and $R^2 = 0.828$ without split.

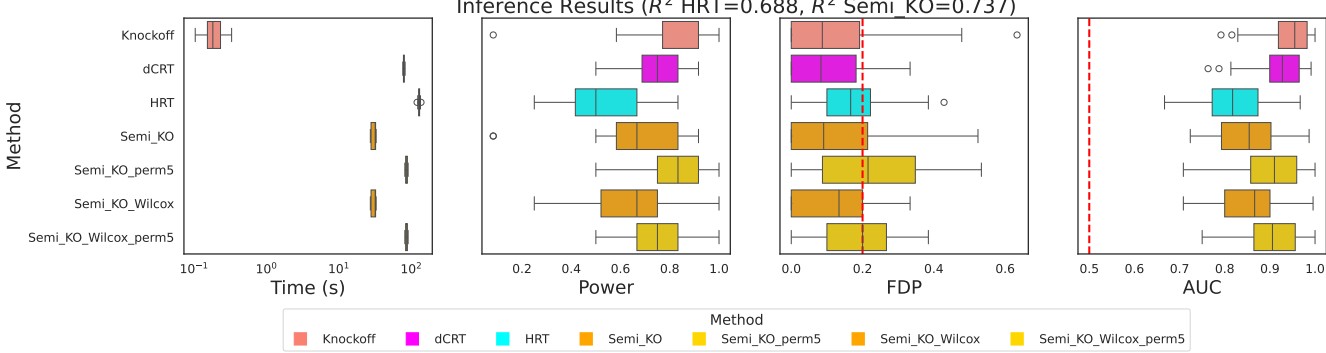

*Figure 23.* **FDR heavy-tailed with Gradient Boosting.** Let $X = \Sigma^{1/2} Z$, where $Z$ has i.i.d. t-Student entries $Z_{ij} \sim t_3$ and $\Sigma_{ij} = 0.6^{|i-j|}$. We generate $y = \beta^\top X + \epsilon$, where the first $0.25p$ coordinates of $\beta$ lie in $[1, 2]$ and the remaining entries are zero, with $\epsilon \sim \mathcal{N}(0, 1)$. The black-box pretrained model $\widehat{m}$ is a gradient boosting, achieving $R^2 = 0.688$ with a train-test split and $R^2 = 0.737$ without split.

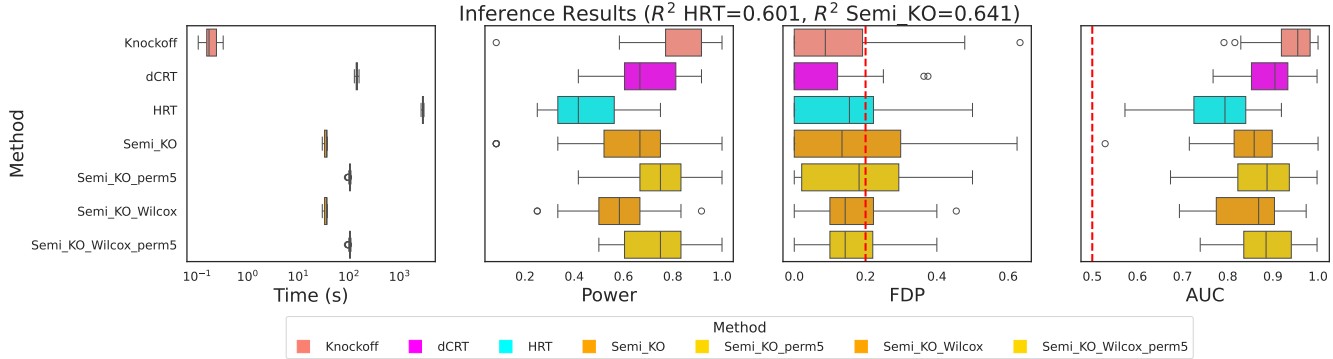

*Figure 24.* **FDR heavy-tailed with Random Forest.** Let $X = \Sigma^{1/2} Z$, where $Z$ has i.i.d. t-Student entries $Z_{ij} \sim t_3$ and $\Sigma_{ij} = 0.6^{|i-j|}$. We generate $y = \beta^\top X + \epsilon$, where the first $0.25p$ coordinates of $\beta$ lie in $[1, 2]$ and the remaining entries are zero, with $\epsilon \sim \mathcal{N}(0, 1)$. The black-box pretrained model $\widehat{m}$ is a random forest, achieving $R^2 = 0.601$ with a train-test split and $R^2 = 0.641$ without split.

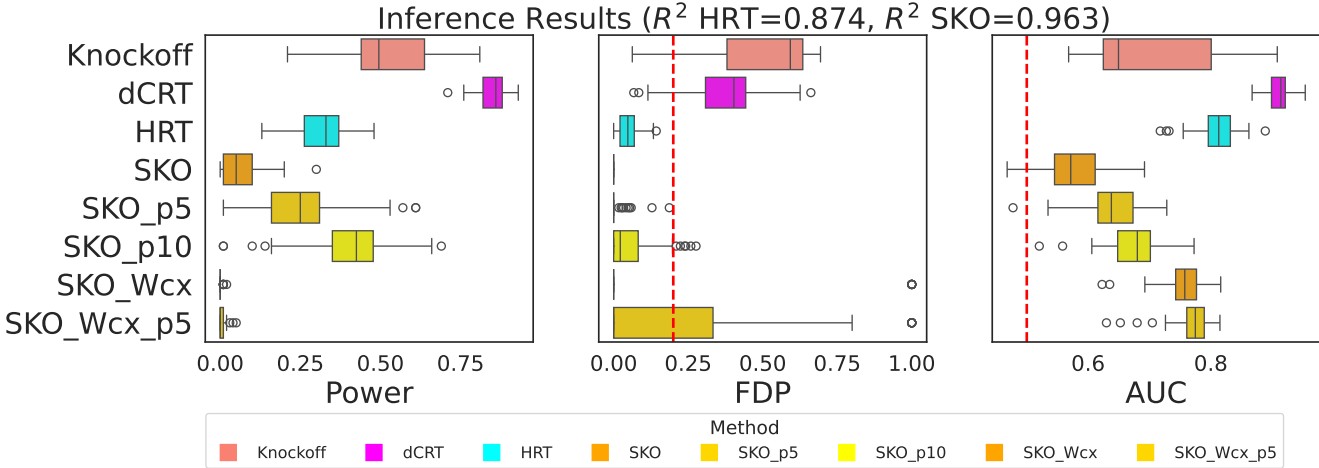

*Figure 25.* **FDR control in High-dimensional setting.** Let $X \sim \mathcal{N}(0, \Sigma)$ with $\Sigma^{ij} = 0.6^{|i-j|}$, and $y = \beta^\top X + \epsilon$, where $0.25p$ coordinates of $\beta$ randomly sparsely sampled from a standard Gaussian and the remaining entries are zero, and $\epsilon \sim \mathcal{N}(0, 1)$. The dimension is $p = 400$ and we use $n = 300$ samples. The black-box pretrained model $\widehat{m}$ is a lasso, achieving $R^2 = 0.874$ with a train-test split and $R^2 = 0.963$ without split.

### F.5.3. HIGH-DIMENSION

In Figure 25, we consider a linear high-dimensional setting with dimension $p = 400$ and sample size $n = 300$. Due to the high dimensionality, we use the lasso both for the learner and for the imputer. The main issue is that, in this regime, estimating the conditional sampler accurately becomes very difficult. As a consequence, the dCRT is unable to provide valid guarantees. Moreover, knockoff sampling, which is intrinsically more challenging than conditional sampling alone, is also inaccurate and knockoffs fails to control the FDR. Such misbehaviors of knockoffs in the presence of correlation and high dimensionality were already observed in Blain et al. (2025).

Nevertheless, we observe that Semi-knockoffs still provide valid guarantees, similarly to the HRT, which is also loss-based. This may be related to the double robustness result. As in the previous experiments, the knockoff threshold used in SKO performs better than the Wilcoxon test followed by BH correction used in SKO-wcx. We also observe a substantial gain in power when using derandomization.

### F.6. Real data experiments

The *Wisconsin Diagnostic Breast Cancer* (WDBC) dataset consists of $n = 569$ patient samples with $p = 30$ numeric features of individual cells, obtained from a minimally invasive fine needle aspirate, to discriminate benign from malignant breast lumps. This dataset is widely used for binary classification tasks in medical informatics (Mangasarian et al., 1995).

Since the underlying distribution is unknown, we estimate the type-I error by adding an artificial null feature correlated at 0.6 with the original inputs. We present results for computation time, number of discoveries, and type-I error using Random Forests, Neural Networks, and Gradient Boosting (Figures 26, 27, and 28, respectively). The conclusions are consistent with those from the simulated settings. The additive term correction is overly restrictive in practice and yields no discoveries (_sqrt). LOCO-W does not control the error. The dCRT makes no discoveries with Neural Networks or Gradient Boosting in the distillation step. Computationally, Semi-knockoffs should provide an advantage since no refitting of the complex model is required; however, in these simulations we used default implementations, so the benefit is less noticeable.

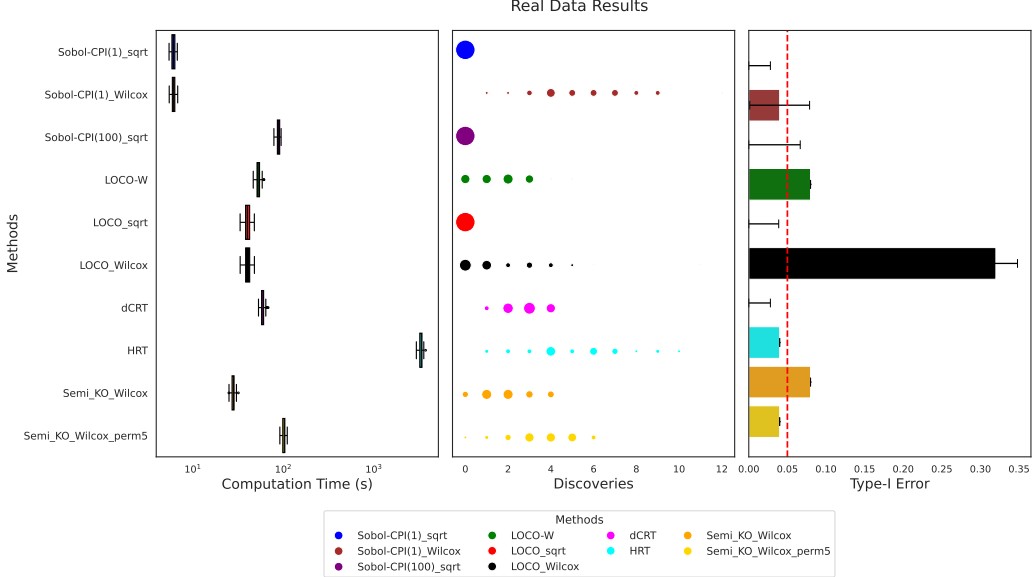

*Figure 26.* **Discoveries on real data using a Random Forest.** Left: computation time; middle: number of discoveries; right: type-I error, estimated using an artificial null feature correlated at 0.6 with the original features. Semi-knockoffs makes discoveries while controlling the error.

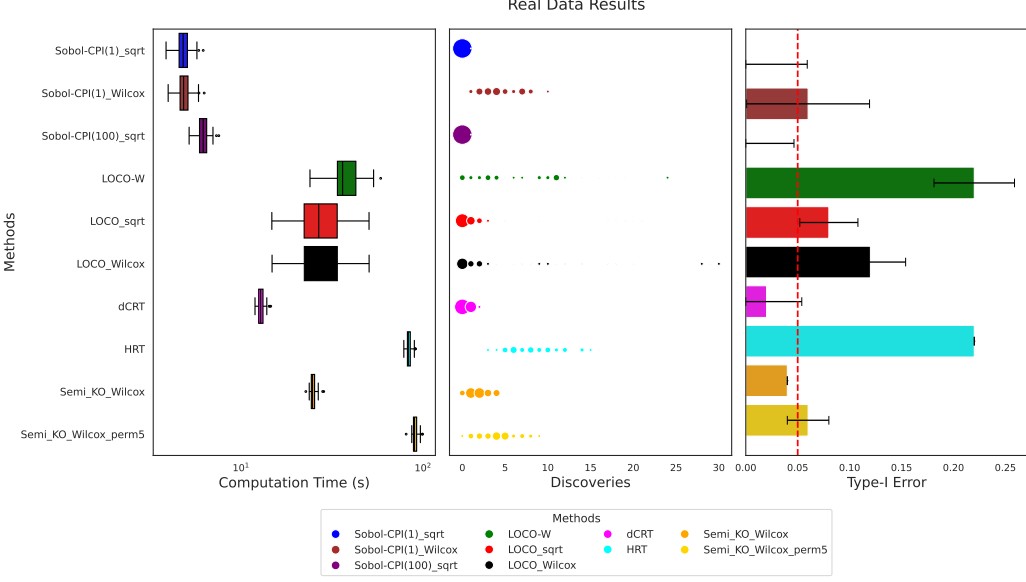

*Figure 27.* **Discoveries on real data using a Neural Network.** Left: computation time; middle: number of discoveries; right: type-I error, estimated using an artificial null feature correlated at 0.6 with the original features. Semi-knockoffs makes discoveries while controlling the error.

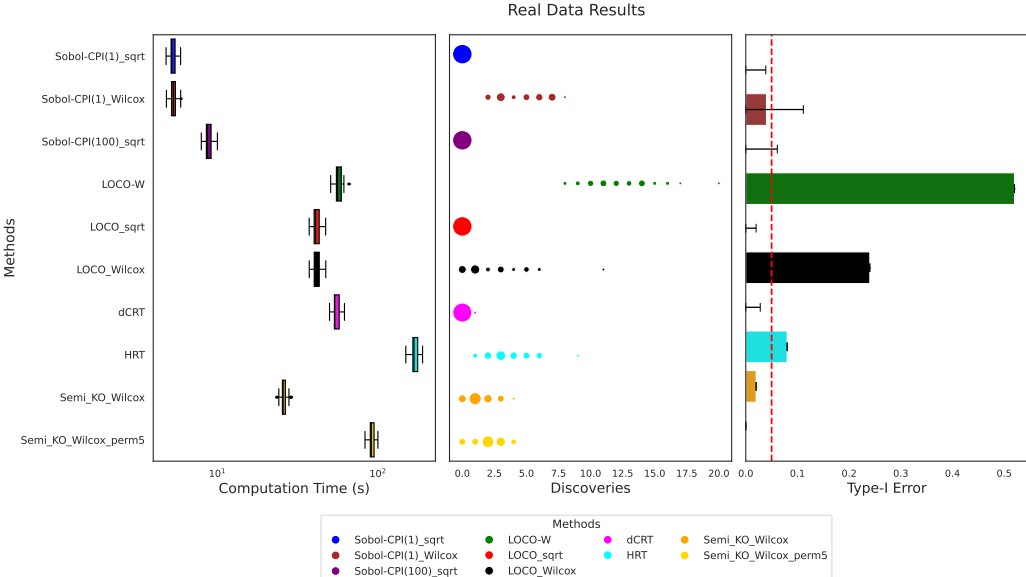

*Figure 28.* **Discoveries on real data using a Gradient Boosting.** Left: computation time; middle: number of discoveries; right: type-I error, estimated using an artificial null feature correlated at 0.6 with the original features. Semi-knockoffs makes discoveries while controlling the error.

