# OpenReview forum: "Semi-knockoffs: a model-agnostic conditional independence testing method with finite-sample guarantees"
_ICML.cc/2026/Conference — ICML 2026 regular_

### Official Review · Reviewer_yFjt · 2026-03-06

**Soundness:** 2
**Presentation:** 3
**Significance:** 3
**Originality:** 3
**Overall Recommendation:** 4
**Confidence:** 2

**Summary:**

The paper suggests a conditional independence testing method that works with arbitrary pre-trained machine learning models while avoiding the need for a train–test split. The authors introduce Semi-knockoffs, which constructs two perturbed versions of each feature using conditional expectation models—one that conditions on the other features and another that also conditions on the other features and target variable. Under the null hypothesis, both perturbations follow the same distribution, which creates a symmetric test statistic that can be used for nonparametric testing and false discovery rate control.
The authors outline a major issue in existing model-agnostic testing approaches, namely that they often require a train–test split to maintain independence between the model and the test statistic, which reduces statistical power. The proposed method aims to avoid this limitation.

**Compliance With Llm Reviewing Policy:**

Affirmed.

**Final Justification:**

Overall, while I am not an expert in this specific area, I find the method conceptually novel and practically promising, and the authors’ clarifications strengthen my confidence in its relevance.

**Key Questions For Authors:**

--

**Limitations:**

yes

**Strengths And Weaknesses:**

Strengths:
- The symmetric perturbation mechanism provides a novel way to avoid train–test splitting while maintaining exchangeability.
- The method can be used with arbitrary pre-trained machine learning models.
- The paper includes several interesting theoretical contributions, including a stability result for regularized empirical risk minimization when adding a null feature and an analysis of distributional convergence under the null hypothesis.
- The experiments compare the proposed method with several baselines from both variable importance and conditional independence testing literature, showing promising performance in simulated settings.

Weaknesses:
- The strongest theoretical guarantees are derived under the assumption that the conditional expectations used to generate the semi-knockoff variables are known exactly. In practice, these quantities must be estimated using regression models. While the paper provides supporting arguments based on stability and distributional convergence, it does not fully establish that the same finite-sample guarantees hold for the practical version with estimated conditional expectations.
- Some parts of the theoretical justification rely on conjectural reasoning. In particular, the argument that the empirical statistic preserves the sign of the oracle statistic with high probability is motivated by a double-robustness intuition rather than a formal proof. This leaves some uncertainty regarding whether the desired exchangeability property is preserved under estimation.
- The experimental evaluation includes several simulated experiments and a small number of real datasets. Additional experiments on larger and more diverse real-world datasets would strengthen the empirical evidence for the method’s robustness and practical usefulness.
- The procedure requires fitting conditional expectation models for each feature and generating multiple perturbations. This may lead to non-negligible computational cost in high-dimensional settings.

---

> ### Author Rebuttal · Authors · 2026-03-29
>
> Thank you for your review and for recognizing the novelty and promising practical impact of our procedure, which can accommodate arbitrary machine learning models.
>
> **Weaknesses:**
>
> 1. It is well known that the null hypothesis of Conditional Independence Testing is too large to admit generally powerful tests without additional assumptions (Shah and Peters, 2020). Therefore, some form of structural assumption is unavoidable. In the standard knockoff framework (Candès et al., 2018), this takes the form of the model-X assumption, which requires the ability to construct knockoff variables satisfying a strong exchangeability property. This is more demanding than conditional sampling alone, as it requires exchangeability with respect to both the original features and the previously constructed knockoff variables. In practice, this property is rarely satisfied exactly, and when it fails, no guarantees hold (Blain et al., 2025).
>
>     In our approach, we shift this burden toward estimating conditional expectations, which is the primary objective of most machine learning models and is therefore more natural and practical than sampling from complex conditional distributions. Throughout the paper, we have been transparent about this non-oracle limitation (which is not fulfilled by the Knockoffs either), while also providing strong intuition for why the method performs well in practice. In particular, we introduce two properties of independent interest: stability and double robustness.
>
> 2. The double robustness property is formally established. What remains conjectural is its direct implication for FDR control. Nevertheless, exchangeability can still be preserved with high probability (see our response to Reviewer CzXy), which supports the validity of the procedure in practice.
>
> 3. We agree that additional experiments on real datasets would further strengthen the empirical validation. As part of future work, we are conducting a more applied study aimed at making Semi-knockoffs fully practical for end users. This includes: analyzing the role of imputers and their connection to the missing data literature (e.g., incorporating outputs into imputations), exploring derandomization via Rao-Blackwellization, extending the framework to sequential conditional independence testing using symmetric e-values (de la Peña, 1999), and applying the method to genomic datasets.
>
>  4. We expect the samplers used in our method to be simpler than the global predictive model, and therefore not to scale as poorly with dimension. Moreover, Conditional Randomization Test (CRT) methods also require a sampler for each feature, typically involving conditional densities, which are more complex than conditional expectations. In addition, generating perturbations in our method is computationally light, as it mainly involves permutations. Finally, the number of resamples required in our approach (typically around 5) is much smaller than in CRT methods, which often require on the order of $10^4$ resamples to obtain accurate $p$-values.
>
> We thank the reviewer for these insightful comments and hope that this addresses their concerns.
>
> 1. Shah, R. and Peters, J. The hardness of conditional independence testing and the generalised covariance measure. Annals of Statistics
> 2. Candès, E., Fan, Y., Janson, L., and Lv, J. Panning for
> gold:‘model-X’knockoffs for high dimensional controlled
> variable selection. Journal of the Royal Statistical Society:
> Series B (Statistical Methodology)
> 3. Blain, A., Lobo, A. R., Linhart, J., Thirion, B., and Neuvial, P. When knockoffs fail: diagnosing and fixing
> non-exchangeability of knockoffs, 202
> 4. De La Peña, V. H. (1999), “A General Class of Exponential Inequalities for Martingales and Ratios”,
> The Annals of Probability

---

> > ### Author Rebuttal · Reviewer_yFjt · 2026-04-02
> >
> > Thank you for the detailed clarification—I appreciate the discussion of assumptions and practical considerations, which helps better position the method, and I maintain my positive assessment.

---

### Official Review · Reviewer_5pSK · 2026-03-08

**Soundness:** 3
**Presentation:** 3
**Significance:** 3
**Originality:** 3
**Overall Recommendation:** 5
**Confidence:** 3

**Summary:**

This paper presents Semi-knockoffs, a conditional independence testing method that avoids using a train-test split and provides type-I error and FDR control without constructing valid knockoff variables.

**Compliance With Llm Reviewing Policy:**

Affirmed.

**Final Justification:**

I increased my score as authors addressed all points, and I believe that this paper has a valuable contribution, especially in terms of practical applications.

**Key Questions For Authors:**

* Could you provide further experiments to study the outcomes under varying sample size ?
* Could you provide a more thorough computational analysis?

**Limitations:**

Yes

**Strengths And Weaknesses:**

Strengths
* The paper is generally well written and the description of the problem is clear.
* The connection to knockoffs is clear, as well as how the paper differentiates itself from knockoffs.
* The paper is original, and the contribution is valuable, as constructing knockoffs is very difficult for large datasets.
* The authors are aware of the limitations and name them clearly in their paper.

Weaknesses
* The double robustness result is a conjecture, not a proven theorem, which seems like a significant gap.
* The stability results applies only to L2 regularized models.
* The experiments use only one sample size.

---

> ### Author Rebuttal · Authors · 2026-03-29
>
> Thank you for your review and for acknowledging the originality and necessity of our approach. In particular, the knockoffs may not behave as expected in finite-sample and high-dimensional settings, which are precisely the regimes for which they were originally introduced.
>
> **On the weaknesses:**
>
> 1. The double robustness result is formally established as a theorem, relying on a Taylor expansion argument. What remains conjectural is its role in ensuring empirical FDR control. Nevertheless, this result can still be leveraged to obtain approximate exchangeability in probability, which supports the practical validity of the procedure.
>
> 2. The stability result indeed relies on $\ell_2$ regularization. However, we have also experimented with other imputers, such as Lasso, and observed similar empirical behavior. While extending this property to arbitrary machine learning models may not be feasible, it remains a desirable feature: the prediction should not vary significantly when a purely noisy feature is introduced.
>
> 3. We agree that this choice is somewhat arbitrary, as discussed below.
>
>
> **Key questions:**
>
>
> 1. The choice of $n=300$ is indeed arbitrary. To assess robustness, we repeated the simulation in the adjacent setting with $p=50$ and varying sample sizes $n \in \{100, 200, 400\}$. The conclusions remain unchanged across these settings. Results are available at https://anonymous.4open.science/r/Semi_KO-6224/results/main_figures/p_values/rebuttal_adjacent_GB_n100.pdf.
>
> 2. Computational aspects are primarily discussed in the appendix (Section F.4), as the procedure could be significantly improved through parallelization for instance. Additionally, distillation strategies, similar to those used in dCRT, could be employed instead of refitting the model with augmented inputs. Overall, we expect the method to be as efficient as SCPI, since the computational bottleneck lies in training the global model rather than the imputers, and the main difference with SCPI is retraining 2 imputers rather than 1. Consequently, it is lighter than LOCO-based approaches. Compared to CRT methods, our approach requires far fewer resamples for derandomization (typically around 5), whereas CRT procedures often require thousands of resamples to obtain accurate $p$-values.
>
>
> Thank you again for your thoughtful review. We hope that this addresses your concerns.

---

> > ### Author Rebuttal · Reviewer_5pSK · 2026-04-01
> >
> > The authors addressed all questions.

---

### Official Review · Reviewer_CzXy · 2026-03-09

**Soundness:** 4
**Presentation:** 4
**Significance:** 4
**Originality:** 4
**Overall Recommendation:** 5
**Confidence:** 3

**Summary:**

This paper proposes Semi-knockoffs, a model-agnostic conditional independence testing framework that avoids train–test splitting and aims to provide finite-sample guarantees for both per-feature p-values and FDR control. The core idea is to construct two conditionally-resampled inputs using regressors for $E[X^j | X^{-j}] and E[X^j | X^{-j}, Y]$, and to compare the losses of a fixed pre-trained model under these two synthetic populations. Under the null, these two synthetic distributions coincide, enabling paired nonparametric tests; meanwhile, for multiple testing, a knockoff-style threshold based on symmetry of signs. The paper further studies the practically relevant case where conditional expectations are estimated, giving a stability result for regularized ERM, a Wasserstein-distance control under the null, and a “double-robustness” style rate result intended to justify approximate validity and power.

**Compliance With Llm Reviewing Policy:**

Affirmed.

**Key Questions For Authors:**

1. For FDR control with estimated imputers, does the knockoff sign-exchangeability lemma still hold?
2. How sensitive is Semi-knockoffs to the choice of imputer class for ν̂ and ρ̂? Do nonlinear imputers (e.g., RF/NN) improve power under alternatives while keeping Type I near nominal under the null?
3. Can you expand high-dimensional (p >> n) experiments, report runtime scaling, or include additional real datasets (such as genotype-phenotype data)?

**Limitations:**

Yes.

**Strengths And Weaknesses:**

Strengths:
1. The authors introduces a novel way to induce exchangeability without constructing full model-X knockoffs: compare two synthetic distributions (condition on X^{-j} vs. on X^{-j}, Y) so symmetry holds under the null even without a train–test split.
2. The Semi-knockoff statistic integrates natively with a pre-trained black-box predictor and allows knockoff-style FDR thresholding via sign symmetry instead of antisymmetric statistics on (X, $\tilde{x}$).
3. The stability result for regularized ERM when adding a null feature is of independent interest; connecting parameter stability to distributional closeness is a thoughtful line of analysis.
4. The paper proposes a important solution to the situation that train–test splits reduce power; Semi-knockoffs proposes a way around this while aiming to retain finite-sample guarantees.

Weaknesses:
1. There is limited coverage of high-dimensional regimes (p >> n) where knockoff idea is typically applied.
2. Finite-sample guarantees appear to be fully established only in the oracle setting from the paper, which requires further elaboration.

---

> ### Author Rebuttal · Authors · 2026-03-29
>
> Thank you very much for your review and for acknowledging the novelty of our approach, as well as its ability to accommodate arbitrary machine learning models while avoiding the statistical loss induced by a train-test split.
>
> **With respect to the weaknesses:**
>
> 1. It is true that knockoffs were originally designed for high-dimensional settings. Moreover, due to the +1 term in the denominator, their threshold can be less effective when the number of truly important features is small. Nevertheless, recent work (Blain et al., 2025) shows that empirical knockoffs may fail to provide FDR guarantees in high-dimensional regimes. In particular, for Gaussian samplers, estimating the inverse of a high-dimensional covariance matrix is challenging. Similarly, alternative samplers such as DeepKnockoffs (Romano et al., 2020) or DeepLink (Zhu et al., 2021) may struggle to accurately learn high-dimensional distributions which is a hard problem. In contrast, we expect Semi-knockoffs to exhibit improved behavior due to their double robustness property. To support this, we conducted an additional high-dimensional simulation (with $p=400$ and $n=300$), available at https://anonymous.4open.science/r/Semi_KO-6224/results/main_figures/KO/rebuttal_hidim2_lasso.pdf, where we observe that knockoffs fail to control the FDR, while Semi-knockoffs maintain control.
>
> 2. In practice, there are no exact finite-sample guarantees for knockoffs. The original works (Barber and Candès, 2015; Candès et al., 2018) provide guarantees only under the assumption that knockoff variables can be sampled exactly. More recent works (Fan et al., 2025) attempt to address estimation error in knockoff construction, but only establish asymptotic FDR control. We agree that obtaining finite-sample guarantees that remain valid in high-dimensional settings is highly desirable. However, it is important to note that making assumptions is unavoidable in Conditional Independence Testing if one seeks non-trivial tests (Shah and Peters, 2020).
>
> **Questions:**
>
> 1. The result does not hold exactly as in the oracle setting, but an approximate version can be established. Specifically, assume that the statistic is absolutely continuous with respect to the Lebesgue measure, with density $f$ satisfying $\|f\|_\infty < \infty$. Then,$\mathbb{P}\bigl(\mathrm{sign}(\widehat{W}) \neq \mathrm{sign}(W)\bigr) \leq O(a_n b_n),$ where $a_n$ and $b_n$ are the convergence rates arising from the double robustness theorem.
>
> 2. From a theoretical perspective, it is unclear whether a stability result holds for a learner that is not explicitly regularized. Asymptotically, such a result should hold if the learner is consistent, which relates to the LOCO (Williamson et al., 2021) converging to zero. However, in finite samples, stability must be analyzed on a model-by-model basis. This notion of stability is appealing: adding a noisy variable should not significantly affect the model. From a practical standpoint, more expressive imputers may capture complex relationships better and thus improve power. Exploring these trade-offs, including applications to genotype data, is an interesting direction for future work.
>
> 3. We expanded our simulations to include a higher-dimensional setting (see https://anonymous.4open.science/r/Semi_KO-6224/results/main_figures/KO/rebuttal_hidim2_lasso.pdf), using the adjacent setup as in the paper but with $p=400$ instead of $p=200$ (with $n=300$ and correlation $0.6$). The conclusions remain consistent: FDR control is maintained, the derandomized version improves power, and applying the knockoff threshold directly yields better FDR control than applying the BH procedure to the Wilcoxon statistics. Runtime scaling is reported in the appendix. We emphasize that our procedure remains computationally competitive, as the complex global model is trained only once, and fewer conditional resampling steps are required compared to CRT-based methods.
>
> We thank the reviewer for these insightful comments and hope that this addresses their concerns.
>
> 1. Blain, A., Lobo, A. R., Linhart, J., Thirion, B., and Neuvial, P. When knockoffs fail: diagnosing and fixing non-exchangeability of knockoffs, 2025
> 2. Romano, Y., Sesia, M., and Candès, E. (2020). Deep knockoffs. Journal of the American
> Statistical Association, 115(532):1861–1872.
> 3. Zhu, Z., Fan, Y., Kong, Y., Lv, J., and Sun, F. (2021). Deeplink: Deep learning inference
> using knockoffs with applications to genomics. Proceedings of the National Academy of
> Sciences
> 4. Shah, R. D. and Peters, J. (2020). The hardness of conditional independence testing and the generalised covariance measure. The Annals of Statistics.
> 5. Fan, Y., Gao, L., Lv, J., and Xu, X. (2025). Asymptotic FDR control with model-X knockoffs: Is moments matching sufficient?
> 6. Williamson, B. D., Gilbert, P. B., Simon, N. R., and Carone,
> M. A general framework for inference on algorithm-
> agnostic variable importance. JASA

---

> > ### Author Rebuttal · Reviewer_CzXy · 2026-04-03
> >
> > The authors addressed all my questions. I maintain a positive evaluation of this work.

---

### Official Review · Reviewer_dCxU · 2026-03-13

**Soundness:** 2
**Presentation:** 3
**Significance:** 3
**Originality:** 3
**Overall Recommendation:** 4
**Confidence:** 4

**Summary:**

This paper introduces Semi-knockoffs, a conditional independence testing (CIT) and variable selection framework which is designed to work with any arbitrary pre-trained predictive model while avoiding the train-test split usually needed in model-agnostic procedures. For each feature $X_j$, the method builds two versions of that feature: one imputed from $E[X_j \mid X_{-j}]$ and one from $E[X_j \mid X_{-j}, Y]$. Under the CIT null $X_j \perp Y \mid X_{-j}$, these two expectations are equal, so any loss-based statistic is symmetric and exchangeability holds. This yields paired-test p-values and motivates an FDR procedure using a knock-off style threshold on the sign-symmetric statistics. In the non-oracle setting, the paper introduces a stability result for regularized ERM and a double-robustness-style argument to explain why the practical procedure where each of the conditional expectations is learned still remains valid and powerful. The authors evaluate their method against a number of baselines on both simulated and real datasets using several black-box models including gradient boosting, neural networks, and random forests.

**Compliance With Llm Reviewing Policy:**

Affirmed.

**Final Justification:**

Final recommendation: weak accept (4)

Using null symmetry to recover knockoff-style sign symmetry without requiring the construction of actual knockoff variables is a clever idea, and the motivation behind this paper is clear. It is reassuring to see that the method retains validity even when the imputers don't align, but sometimes power suffers in the experiments, slightly dampening enthusiasm for the proposed method.

**Key Questions For Authors:**

1. Can the authors prove the conjecture regarding exchangeability under the null for the practical, non-oracle version? It would be helpful to have a more formal statement of this result.

2. Is it possible to provide a more formal statement connecting Wasserstein 1 closeness or double robustness to the rejection rule (i.e. how do Theorem 4.1-4.3 justify the practical behavior of Algorithm 4 when the oracle conditional expectations are not known)?

3. How does the method  scale computationally when p is large?

4. What happens when imputers are misspecified? Is it possible to perform ablations or provide more details on the quality of the imputers relative to the performance of the method?

**Limitations:**

Yes

**Strengths And Weaknesses:**

**Strengths**

1. The oracle setting results are technically sound and the new stability theorem is a good effort to try to bridge the gap between the oracle and estimated case.

2. The motivation is well-explained, and the core construction is easy to follow.

3. Using null symmetry to recover knockoff-style sign symmetry without requiring the construction of actual knockoff variables is clever.

4. Model agnostic CIT and FDR control without sample splitting is an important and potentially impactful area of research.

**Weaknesses**

1. The finite-sample guarantees for CIT claimed in the paper really only apply to the oracle setting where conditional expectations are perfectly known.

2. In the practical regime where conditional expectation functions are estimated, the authors rely on a stability result and conjecture exchangeability under the null (signs are Rademacher) based on the double robustness argument, without providing a formal proof of this.

3. I find the assumptions in the paper to be somewhat narrower than the framing suggests, though the authors do acknowledge that some of the formal smoothness assumptions may be stronger than what is needed in practical. For example, the double-robustness result (Theorem 4.3) requires differentiable fitted predictors, but the authors claim (based on heuristics and empirical results) that this result can still hold for non-smooth learners like random forests. Still, there seems to be a gap between the theory and empirical results.

4. In high-dimensional settings, learning the conditional expectation functions could be challenging and in non-linear settings, if $\hat{\rho}$ and $\hat{\nu}$ differ for reasons that are not related to the null, then the method may suffer from miscalibration which is not addressed in the theory. It feels like these estimates might suffer in finite sample from severe bias without imposing any restrictions/assumptions which could compromise validity.

5. This is minor, but there are some language imprecisions: for example, “both their predictions and the residuals from which we sample are similar, so the two sampling schemes yield observations that are iid” is not quite precise.

---

> ### Author Rebuttal · Authors · 2026-03-29
>
> Thank you for your review and for highlighting our efforts in avoiding a potentially impactful train-test split. This is indeed a common problem in Machine Learning, particularly in Conditional Independence Testing, where there is a trade-off between using complex machine learning methods and leveraging all available data to achieve statistical accuracy. Therefore, we expect that the new direction introduced in this paper will help address this important practical finite-sample issue.
>
> **With respect to the weaknesses:**
>
> 1. Yes, the exact recovery results hold only when the conditional expectation is known exactly. However, having guarantees and power without reducing the null hypothesis set is unavoidable (Shah \& Peters, 2020). For this reason, we introduced this assumption, which appears more practical in a machine learning context, as it relies on estimating a prediction function rather than a full conditional distribution, which is intrinsically more challenging. Moreover, standard knockoffs (Barber and Candès, 2015; Candès et al., 2018) rely on the ability to construct knockoff variables by iteratively sampling from conditional distributions given other input features and previously constructed knockoffs ($\tilde{X}^j \sim \mathcal{L}(X^j \mid X^{-j}, \tilde{X}^1, \ldots, \tilde{X}^{j-1})$, SCIP algorithm). Also, there is a potential cumulative error that does no appear in our method that does not depend on the other estimations. Without exact construction, no control is guaranteed. While some recent asymptotic results exist (Fan et al., 2025), negative empirical results on finite-sample control have also been reported (Blain et al., 2025) when variables are estimated.
>
> 2. Even though the formal link between double robustness and the conjectured FDR control is not yet established, we expect that this property improves finite-sample FDR control compared to standard knockoffs. Indeed, in standard knockoffs, the importance of a null feature tends to zero because the model does not assign it importance. In our method, not only does the model identify the feature as unimportant, but the imputer also contributes to this effect.
>
> 3. This proof relies on the fact that, under the conditional null, the model does not depend on the feature since it is not predictive. Formalizing this independence for general machine learning models is challenging. The differentiability assumption arises from the use of a Taylor expansion in our proof. Alternatively, one could assume an asymptotic independence condition of the model from the unimportant features sufficiently smooth to recover rates from the imputers.
>
> 4. Providing guarantees in high-dimensional settings is indeed challenging, although this is a key motivation for knockoffs. For instance, Gaussian knockoffs can fail in high dimensions due to difficulties in estimating the inverse covariance matrix (Blain et al., 2025). We expect our method to scale better with dimension due to its double robustness property, as supported by our additional experiments: (n=300 and p=400) https://anonymous.4open.science/r/Semi_KO-6224/results/main_figures/KO/rebuttal_hidim2_lasso.pdf.
>
> 5. We thank you for pointing out the language imprecisions.
>
> **Questions:**
>
> 1. 2. As discussed above, this is an intrinsically difficult problem that remains unsolved even for standard knockoffs. Nevertheless, we have extended our theoretical analysis and obtained two new results. First, we establish FDR control by iteratively bounding the gap with the theoretical quantity and applying knockoff control to the latter. This introduces a term $|\mathcal{H}_0|/\sqrt{n}$, leading to asymptotic control but not exact finite-sample control in high dimensions. We expect this bound to improve when incorporating double robustness arguments. Second, for the SKO-Wilcoxon procedure, we derive a type-I error control involving an additive total variation term, similar to Berret et al. (2020), along with a bound in Kolmogorov distance rather than Wasserstein distance. Bridging this remaining gap is part of future work.
>
> 3. The complex learner is trained only once. As dimensionality increases, computational cost remains manageable since simpler models are used for the imputers.
>
> 4. In our experiments, even with an inaccurate sampler, the method does not severely violate FDR control. This is because the model tends to assign low importance to irrelevant features, and the samplers often share similar inaccuracies, leading to similar expectations. However, this reduces statistical power.
>
>
> Thank you again for your insightful feedback. We fully understand your concerns and have aimed to be clear and transparent throughout the paper regarding both results and limitations (as is also the case for standard knockoffs, although this is not always widely recognized). Despite the remaining gaps, we believe our contribution is both novel and significant. We would be happy to address any further questions.

---

> > ### Author Rebuttal · Reviewer_dCxU · 2026-04-03
> >
> > Thank you for the clarifications and detailed responses. In your attached experiment, it seems HRT performs the best out of all methods. Can you comment on this?

---

> > > ### Author Response · Authors · 2026-04-07
> > >
> > > Thank you very much for your interest and for the follow-up.
> > >
> > > ***FDR control only for oracle knockoffs, not empirical ones***
> > >
> > > The goal of that figure was not to demonstrate increased power by avoiding the train–test split (this is achieved in this high-dimensional setting by increasing the number of permutations as discussed below). Rather, it was intended to show that, contrary to expectations from the knockoffs literature, where knockoffs are introduced as a finite-sample FDR-valid procedure suitable for high-dimensional settings, this guarantee does not always hold in practice. This limitation arises because knockoff variables are difficult to sample accurately.
> > >
> > > Nevertheless, we observe that our method remains valid. We therefore emphasize that obtaining guarantees for empirical Conditional Independence Testing procedures is challenging and, to the best of our knowledge, has not yet been fully achieved, even for knockoffs. Our work goes beyond proposing an oracle procedure: we explicitly discuss whether and under what conditions the empirical version potentially preserves these guarantees, and we provide supporting arguments toward this goal.
> > >
> > > ***Gain in power from avoiding the train–test split***
> > >
> > > To illustrate the gain in power from avoiding the train–test split compared to the HRT, we refer to the figures in the main text, as well as Figures 14–18 (type-I error) and Figures 20–24 (FDR).
> > >
> > > We also repeated the high-dimensional experiments by incorporating permutations into the derandomization step (SKO_p10 with 10 permutations). The results show that our method remains more powerful than the HRT:
> > > https://anonymous.4open.science/r/Semi_KO-6224/results/main_figures/KO/rebuttal_hidim2_lasso.pdf
> > >
> > > Additionally, we confirmed that the same conclusions hold in the extra experiments requested by Reviewer 5pSK, across different sample sizes (n = 100, 200, 400) and models (GB, RF, NN). See, for example:
> > > https://anonymous.4open.science/r/Semi_KO-6224/results/main_figures/p_values/rebuttal_adjacent_GB_n100.pdf
> > >
> > > All related experiments are available in the same folder.
> > >
> > > Overall, there is a significant gain in discovery power when using all available samples. This advantage is especially pronounced in low-sample regimes (e.g., n = 100).
> > >
> > > We thank the reviewer for their insightful comments and hope this clarifies and addresses their concerns.

---

### Decision · Program_Chairs · 2026-04-30

**Decision:**

Accept (regular)

**Comment:**

This submission is recommended for Acceptance based on the reviewers' consensus that its "clever" use of null symmetry provides a novel, model-agnostic way to achieve FDR control without the power loss of sample splitting. The authors' rebuttal successfully addressed technical concerns by providing new high-dimensional simulations where the method outperformed standard knockoffs and by establishing new results for approximate exchangeability under estimation. Ultimately, the reviewers agree that the work is technically solid and offers a practically impactful solution for statistical inference with black-box models.